# Modeling impacts of faster productivity growth to inform the CGIAR initiative on Crops to End Hunger

Keith Wiebe[1]*, Timothy B. Sulser[1], Shahnila Dunston[1], Mark W. Rosegrant[1], Keith Fuglie[2], Dirk Willenbockel[3], Gerald C. Nelson[4]

1 International Food Policy Research Institute, Washington, DC, United States of America, 2 United States Department of Agriculture, Economic Research Service, Washington, DC, United States of America, 3 Institute of Development Studies, University of Sussex, Brighton, United Kingdom, 4 University of Illinois, Urbana-Champaign, IL, United States of America

* k.wiebe@cgiar.org

## Abstract

In 2017–2018, a group of international development funding agencies launched the Crops to End Hunger initiative to modernize public plant breeding in lower-income countries. To inform that initiative, USAID asked the International Food Policy Research Institute and the United States Department of Agriculture's Economic Research Service to estimate the impacts of faster productivity growth for 20 food crops on income and other indicators in 106 countries in developing regions in 2030. We first estimated the value of production in 2015 for each crop using data from FAO. We then used the IMPACT and GLOBE economic models to estimate changes in the value of production and changes in economy-wide income under scenarios of faster crop productivity growth, assuming that increased investment will raise annual rates of yield growth by 25% above baseline growth rates over the period 2015–2030. We found that faster productivity growth in rice, wheat and maize increased economy-wide income in the selected countries in 2030 by 59 billion USD, 27 billion USD and 21 billion USD respectively, followed by banana and yams with increases of 9 billion USD each. While these amounts represent small shares of total GDP, they are 2–15 times current public R&D spending on food crops in developing countries. Income increased most in South Asia and Sub-Saharan Africa. Faster productivity growth in rice and wheat reduced the population at risk of hunger by 11 million people and 6 million people respectively, followed by plantain and cassava with reductions of about 2 million people each. Changes in adequacy ratios were relatively large for carbohydrates (already in surplus) and relatively small for micronutrients. In general, we found that impacts of faster productivity growth vary widely across crops, regions and outcome indicators, highlighting the importance of identifying the potentially diverse objectives of different decision makers and recognizing possible tradeoffs between objectives.

**Data Availability Statement:** Data are available on GitHub at https://github.com/IFPRI/IMPACT.

**Funding:** KW, TBS, SD, and MWR received funding from the United States Agency for International

Development (USAID; https://www.usaid.gov). DW
and GCN received funding from the International
Food Policy Research Institute (IFPRI; https://
www.ifpri.org/). Funding for model development
and maintenance was provided by the CGIAR
Research Program on Policies, Institutions, and
Markets (PIM; http://pim.cgiar.org/). USAID
participated in the selection of crops and countries
to be studied and the identification of scenarios to
be modeled.

**Competing interests:** The authors have declared
that no competing interests exist.

## Introduction

The world's food systems face the challenge of meeting demands for food commodities that
are projected to rise by 50% or more by mid-century [1–4], even as climate change slows yield
growth for many crops and regions [5–8]. Achieving the Sustainable Development Goals and
other policy objectives will require going beyond meeting food demand to eliminating poverty
and hunger, improving nutrition and health, and reducing environmental impacts. Achieving
these multiple goals will require multiple approaches, including dietary change [9], reductions
in food losses and waste [10, 11], and improvements in agricultural productivity. Productivity
growth has been key to increasing food production over the past half century and will be even
more important in meeting these broader challenges in the future [12–14]. Sources of on-farm
productivity growth include adoption of new varieties, improved inputs, and better manage-
ment techniques. Increased investment in agricultural research and knowledge transfer to
farmers will play a critical role, particularly in developing countries.

In 2017–2018, a group of international development funding agencies, including the United
States Agency for International Development (USAID), the Bill & Melinda Gates Foundation
(BMGF), the UK Department for International Development (DFID), the German Federal
Ministry for Economic Cooperation and Development (BMZ) and the Australian Centre for
International Agricultural Research (ACIAR), launched a program to modernize public plant
breeding in lower-income countries. The Crops to End Hunger (CtEH) initiative seeks to
"accelerate and modernize the development, delivery and widescale use of a steady stream of
new crop varieties. . . for the staple crops most important to smallholder farmers and poor
consumers" [15]. To inform that initiative, USAID asked the International Food Policy
Research Institute (IFPRI) and the United States Department of Agriculture's Economic
Research Service (ERS) to estimate the impacts of faster productivity growth for selected food
crops on income and other key indicators in developing countries in 2030.

## Approach

Fig 1 gives an overview of the approach used to derive estimates of potential impacts of accel-
erated yield growth in target crop commodities. We first prepared estimates of the total value
of production for each crop in each of the 106 countries for a "parity model" analysis [16]
using data from FAOSTAT [17]. We then used IFPRI's International Model for Policy Analy-
sis of Agricultural Commodities and Trade (IMPACT) [18, 19] and the GLOBE general equi-
librium model [20] to estimate changes in the total value of production of those crops to 2030
in the reference case, as well as changes in economy-wide income (or economic surplus) that
would result under scenarios of faster crop productivity growth. The scenarios of accelerated
productivity growth reported in this paper explore the impacts of a hypothetical 25% increase
in the annual rate of yield growth above "baseline" yield growth in farmers' fields over the
period 2015–2030. (This could result from increased investment in new crop varieties or from
other sources of on-farm productivity growth, but we did not analyze the source of the acceler-
ation.) Potential impacts on poverty were determined by weighting the estimates of production
value and income by the extent and depth of poverty in each country. Scenario results from
IMPACT were also used to estimate potential impacts on hunger and selected indicators of
nutrient adequacy.

In consultation with USAID and experts associated with the CtEH initiative, we selected as
the focus of our analysis 20 CGIAR mandate crops, including cereals, root crops and legumes
(Table 1), in 106 countries–including most countries in Africa, Asia, and Latin America except
for China, Brazil, and southern cone countries of South America. Brazil and China are both
large, upper-middle-income countries; Southern Cone countries (Chile, Argentina, Paraguay

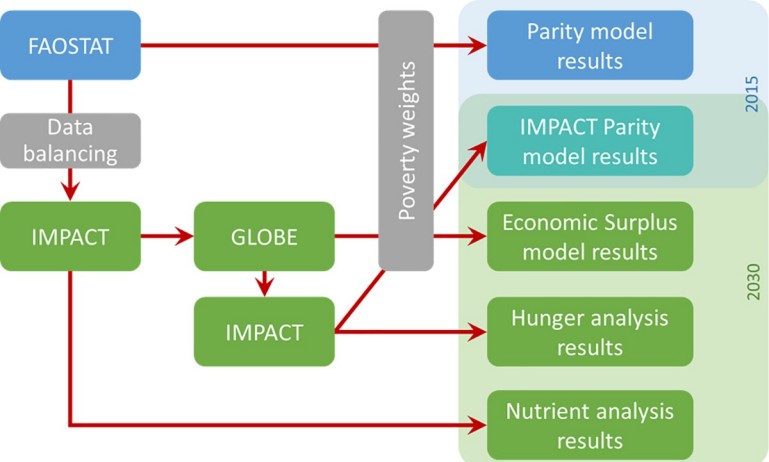

**Fig 1. Summary of methods used.** Source: The authors.

and Uruguay) are middle- or high-income countries that lie mostly in temperate areas. Therefore, these countries are not a primary focus of the CGIAR (see **Table 2** and **Fig 2** for the countries included and regional definitions).

## Analysis using the parity model

Parity or congruence models have often been used as guides or starting points for deciding how to allocate research resources in multi-commodity systems [16, 22, 23]. The "congruence rule" allocates research expenditures across commodities in proportion to each commodity's contribution to the total value of agricultural products. Under the assumptions that, first, opportunities for productivity scientific effort are equivalent in each commodity, and second, the value of a scientific or technical innovation is proportional to the value of the commodity, then an efficient allocation of research expenditures (one that maximizes returns to research) would imply that the research intensity (the ratio of production value to research expenditure) for all commodities should be the same [16]. In the absence of complete and consistent data on the current allocation of research, a parity ratio–the value share of a commodity in the total value of the commodities in question–provides equivalent information, so setting the research expenditure share equal to the value share would assure that the research intensities are equal. Ruttan [16] noted that the congruence or parity rule by itself is insufficient information for an optimal allocation of research resources, as neither of the two underlying assumptions noted above are likely to be true in practice. He did stress, however, that parity rules provide a solid starting point for research resource allocation and that an explicit justification should be developed for any departure from a parity rationale.

One justification for a departure from parity could be distributional or equity concerns. Rather than maximizing the total economic benefits from research, one could focus on the

Table 1. Crops included in the analysis.

| Cereal grains | Barley, maize, millet, rice, sorghum, wheat |
|---|---|
| Roots, tubers & bananas | Banana, cassava, plantain, potato, sweet potato, yams |
| Oilseeds & pulses | Beans, chickpea, cowpea, groundnuts, lentil, pulses (aggregate), pigeonpea, soybean, other pulses |

**Table 2. Countries and regions included in the analysis.**

| | |
|---|---|
| **Latin America & Caribbean** | Belize, Bolivia, Colombia, Costa Rica, Cuba, Dominican Republic, Ecuador, El Salvador, Guatemala, the Guyanas, Haiti, Honduras, Jamaica, Mexico, Nicaragua, Panama, Peru, Venezuela, Other Caribbean |
| **Sub-Saharan Africa** | **Central**: Cameroon, Central African Republic, Chad, Congo, DR Congo, Equatorial Guinea, Gabon |
| | **Eastern**: Burundi, Djibouti, Eritrea, Ethiopia, Kenya, Rwanda, Somalia, Sudan and South Sudan, Uganda |
| | **Southern**: Angola, Botswana, Lesotho, Madagascar, Malawi, Mozambique, Namibia, South Africa, Swaziland, Tanzania, Zambia, Zimbabwe |
| | **Western**: Benin, Burkina Faso, Côte d'Ivoire, Gambia, Ghana, Guinea, Guinea-Bissau, Liberia, Mali, Niger, Nigeria, Senegal, Sierra Leone, Togo |
| **West Asia & North Africa** | Algeria, Egypt, Iran, Iraq, Israel, Jordan, Lebanon, Libya, Mauritania, Morocco, Palestine, Saudi Arabia, Syria, Tunisia, Turkey, Yemen, Rest of Arabia |
| **Central Asia** | Armenia, Azerbaijan, Georgia, Kyrgyzstan, Tajikistan, Turkmenistan, Uzbekistan |
| **South Asia** | Afghanistan, Bangladesh, Bhutan, India, Sri Lanka, Nepal, Pakistan |
| **Southeast Asia** | Cambodia, Fiji, Indonesia, Laos, Malaysia, Myanmar, Papua New Guinea, Philippines, Solomon Islands, Thailand, Timor Leste, Viet Nam, Vanuatu, Other Southeast Asia |

benefits likely to be gained by certain target populations, such as those whose incomes fall below a poverty line. "Weighted" models would give higher weight to benefits enjoyed by these target groups (or, equivalently, by discounting the value of benefits going to non-target groups). Another justification for a departure from parity could be to address nutritional concerns. To the extent that nutrient quality is not fully valued in commodity prices, the parity rule could lead to overinvestment in quantity- or calorie-based outcomes.

While a precise analysis of the welfare effects is not possible given the data available, we can draw some general conclusions by looking at the extent of poverty in the countries where significant economic impact was achieved. We premise this approach on the assumption that the

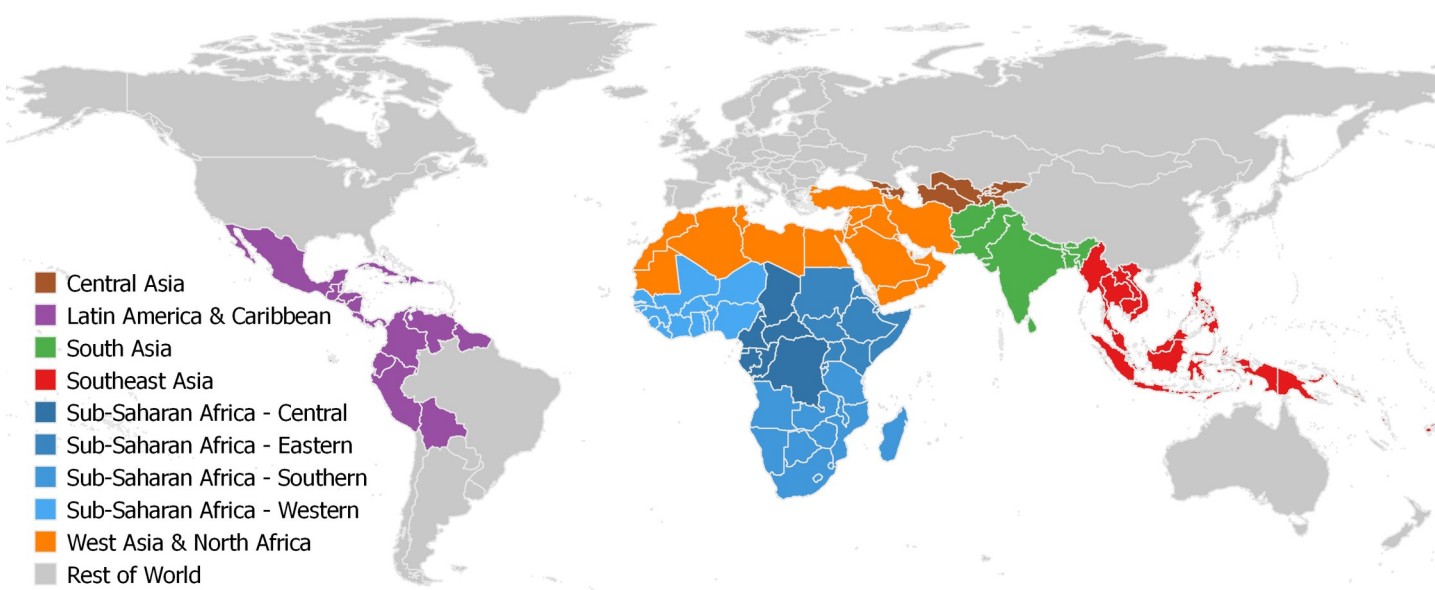

**Fig 2. Countries and regions included in the analysis.** The boundaries and names shown and the designations used on this map do not imply official endorsement or acceptance by the International Food Policy Research Institute (IFPRI). Source: The authors, using an adapted world country boundary map in the ArcWorld Supplement from ESRI, DeLorme Publishing Company, Inc. and created for display here using the free and open source QGIS [21].

economic benefits from accelerated productivity growth in crop staples are widely shared across income classes within a country. Technology adoption raises incomes of farm adopters and, through market-level effects, reduces prices paid by consumers for food. The crops affected are produced predominantly by small-holders, who tend to cluster at the lower end of the income scale. For consumers, income elasticities for these food staples are likely to be positive but small, meaning that per capita levels of consumption do not vary much across income classes (and, as a percentage of total expenditure, are higher for poor households). Thus it is reasonable to assume that benefits are roughly evenly distributed across the income strata of a country, and therefore the share of benefits accruing to those below a poverty line will be correlated with the poverty headcount index for that country at the time these impacts occurred.

In this exercise, the parity model is applied to CGIAR crop commodities using the gross value of commodity production from FAO. This is the total quantity of production averaged over 2014–2016 and valued at global average commodity prices averaged over 2004–2006 (i.e., in constant 2005 international dollars) as reported in FAOSTAT [17]. (These were the latest available at the time the study was done, and they are also consistent with IMPACT's base year of 2005.) The advantage of using this set of prices is that they provide a standardized, revenue-weighted set of commodity prices, expressed in purchasing-power-parity dollars per ton, which value quantities of crops produced around the world in a consistent way. FAO derives this set of prices using the Geary-Khamis method applied to national producer prices from around the world and uses them to construct its index of "Gross Production Value" of every crop in every country in constant international dollars. Prices are based on fresh weight, i.e., using the same measure that FAO uses to report the quantity of production. (We back out these prices by dividing the FAO Gross Production Value by total quantity produced in a year.) (Although FAO recently updated its Gross Production Value index using prices from 2014–2016, a comparison of 2014–2016 prices with 2004–2006 prices reveals that relative crop prices–the value of a ton of wheat relative to a ton of beans, for example–remained quite stable between the two periods despite fluctuations in some years. Thus, while using more recent prices would shift the nominal value of all crops upward, it would have little or no effect on relative values of production or productivity changes across crops, determination of which is the primary objective of this study.)

We applied three weighting schemes to the value of the commodities:

1. The gross value of commodity production is summed across all countries;

2. The gross value of commodity production in each country is first multiplied by the country's $1.90/day/capita poverty rate [24], then summed across countries;

3. The gross value of commodity production in each county is multiplied by the country's $1.90/day/capita poverty rate and its $1.90/day/capita poverty gap [24], then summed across countries.

The weights given to the value of production in (2) and (3) are based on the Foster-Greer-Thorbecke [25] poverty weights that have been widely used in social welfare analysis (although data coverage varies across countries and years). Assuming that per capita consumption of a commodity is roughly equivalent among poor and non-poor, measure (2) essentially only counts commodity consumption by those living below the poverty line. Measure (3) also only counts consumption by these poor but gives higher weight to consumption of the very poor–those living further below the poverty line.

A second departure from the standard parity model is to use projected future values of commodity production rather than current production. Over time, growth in commodity production and utilization are likely to be uneven, as consumer demands change to include more

diversified diets such as meat products and processed foods. Since research investments may take a decade or more to achieve their full impact in farmers' fields, it might be preferable to base today's research and development (R&D) investments on how crop production and utilization are expected to evolve in the future. We used IMPACT to model the value of production of each commodity in each country in 2015 (as a consistency check with FAO data) and to estimate how it might change in 2030 under baseline or "business as usual" assumptions about future socioeconomic and climate change.

## Scenarios of faster productivity growth

A third departure from the simple parity model is to explore how scenarios of accelerated crop yield growth might affect future incomes. While parity calculations are based on commodity production, the IMPACT model also explores how commodities are used and consumed. Faster productivity growth can result in lower prices and wider utilization and trade of commodities in food systems. Importantly, through international trade and price changes, productivity growth in one country can affect (positively or negatively) income in another country. Building on earlier analysis of the impacts of agricultural R&D investments on productivity [26], we used IMPACT to explore how faster productivity growth in each crop might affect future incomes in each of the 106 countries, as a separate and complementary approach to the parity analysis of current and future baseline values of production. For each crop in turn, we ran a scenario in which the baseline rates of productivity growth assumed in the IMPACT model were increased by 25% in the 106 focus countries. For example, if the baseline annual growth rate for rainfed maize yields in a particular country and year was 1.0%, that growth rate was increased to 1.0% x 1.25 = 1.25% per year in the productivity enhancement scenario. Baseline productivity growth rates in IMPACT (available at https://github.com/IFPRI/IMPACT) vary by crop, country, and year (the global weighted average is between 0.7% and 2.2% per year), and each was adjusted accordingly (for both rainfed and irrigated areas) in these scenarios. Yield increases were applied to one commodity at a time, holding other crops to their baseline rate of yield growth. Given that agricultural total factor productivity growth rates in developing countries have averaged about 2% per year since the 1990s [27], a 25% increase in the rate of growth seems like an attainable goal with increased investment in agricultural research and knowledge transfer to farmers.

The simulations thus provide a measure of the potential impact on national income of a similar yield shock applied to each of the 20 commodities. Estimates of income changes were also weighted by the country poverty indices to give greater importance to income gains in countries with larger concentrations of poor people. These income and poverty effects were then aggregated across regions. A data limitation is that we do not have estimates of future poverty rates to weight future income gains, so our best approximation is to assume that present poverty rates are likely to persist for the next decade or so.

## Analysis using the IMPACT model

IMPACT is an integrated system of models linking climate, water, and crop models with a partial equilibrium, multi-market economic model [18, 19]. IMPACT uses assumptions about key drivers such as population, income, technology, policy and climate (described in the next paragraph) to simulate changes in agricultural demand, production and markets for 60 commodities in 158 countries to 2050 (and for intervening years). IMPACT benefits from close interactions with scientists at all 15 CGIAR research centers through the Global Futures and Strategic Foresight (GFSF) program [28, 29], and with other leading global economic

modeling efforts around the world through the Agricultural Model Intercomparison and Improvement Project (AgMIP) [30–32].

In this study the productivity enhancement scenarios for each crop were first run in IMPACT to derive a set of preliminary changes (relative to the baseline) in crop prices, quantities supplied and demanded, harvested areas, and trade for all countries to 2030. Each scenario assumes changes in population and income according to the "Shared Socioeconomic Pathway 2" (SSP 2) [33, 34], which is widely used by global modeling groups as a "business as usual" scenario, and changes in climate based on rapid growth in greenhouse gas emissions according to "Representative Concentration Pathway 8.5" (RCP 8.5) [35–37].

The crop yield changes simulated by IMPACT were passed to the GLOBE general equilibrium model [20] to estimate their broader economy-wide effects. The effects on aggregate household income generated by GLOBE were then passed back to IMPACT to assess the resulting changes in food demand and the associated final changes in prices, supply, area harvested and trade.

## Analysis using the GLOBE model

To capture the broader economy-wide effects of changes in crop productivity, this analysis used an extended dynamic version of the GLOBE model originally developed by McDonald, Thierfelder and Robinson [38]. The model consists of a set of individual region blocs that together provide complete coverage of the global economy and that are linked through international trade and capital flows. Each region bloc represents the whole economy of that region at a sectorally disaggregated level. All sectors are considered simultaneously and the model takes consistent account of economy-wide resource constraints assuming full employment of all resources, intermediate input-output linkages and interactions between markets for goods and services on the one hand and primary factor markets including labor markets on the other. The model simulates the full circular flow of income in each region from (i) income generation through productive activity, to (ii) the primary distribution of that income to workers, owners of productive capital, and recipients of land and other natural resource rents, to (iii) the redistribution of that income through taxes and transfers, and to (iv) the use of that income for consumption and investment. The model version used for the present study is calibrated to the GTAP 9 database [39] and distinguishes 22 production sectors and 15 regions. A detailed description of the model is provided in Willenbockel et al. [20].

The dynamic baseline of GLOBE exactly replicates the aggregated GDP, population and agricultural land supply growth rates as well as the supply price projections for linked agricultural commodity groups of the IMPACT baseline scenario. Moreover, the GLOBE household demand system is calibrated such that the GLOBE income elasticities of demand for food commodities are consistent with the corresponding aggregated IMPACT elasticities [20, 40].

The agricultural productivity enhancements from the various scenarios simulated in IMPACT enter the GLOBE model in the form of shifts of the total factor productivity parameters in the agricultural production functions for the target regions. These productivity shifts affect aggregate household income primarily via their impact on wages, capital and land returns [41]. Employment in the target sectors declines marginally in response to the rise in productivity, as less labor and capital is required than before to satisfy the demand for the targeted crops, given that crop demand is relatively price- and income-inelastic. In economic terms, the drop in the price of crops relative to non-agricultural goods pulls labor and capital from crop production to non-agricultural production–a beneficial side effect from an economic development perspective. This analysis therefore generates projections of the impacts of

agricultural productivity growth on economy-wide household income and GDP in addition to the direct impacts in the agricultural sector.

## Analysis of hunger and selected nutrient indicators

While economic metrics such as income or economic surplus have often been used to evaluate research priorities, other metrics are needed for various types of malnutrition. A final departure from the simple parity model is to consider such metrics. We estimated the number and percentage of children under five years of age who are undernourished based on projections of per-capita calorie consumption from IMPACT combined with assumptions about trends in female access to secondary education, the quality of healthcare, schooling, and access to clean water, using coefficients from Smith and Haddad [42]. We estimated the prevalence of hunger (i.e. the share of the total population at risk of hunger) based on an empirical relationship between the availability of food and the minimum food requirement for each country adapted from Fischer et al. [43].

We also estimated the impact of the productivity enhancement scenarios on availability and adequacy of key nutrients. Nutrient availability is based on the availability of commodities for food (i.e. after excluding animal feed, industrial and other uses, and accounting for imports and exports). With regard to adequacy, medical researchers and health organizations around the world have developed recommendations for needed intake of macro and micronutrients. For this report, we used the U.S. Recommended Dietary Allowance (RDA), the minimum average daily intake of a nutrient needed for the maintenance of good health, as estimated by the Food and Nutrition Board of the U.S. National Academies of Sciences, Engineering, and Medicine [44]. The RDA varies by age and gender and for women who are pregnant or lactating. The Food and Nutrition Board reports RDAs for three macronutrients, 15 minerals and 14 vitamins and other organic micronutrients [44]. We focused on the following:

- Macronutrients (3)–carbohydrates, protein, total fiber

- Minerals (6)–calcium, iron, magnesium, phosphorus, potassium, and zinc

- Vitamins (9)–Vitamin A RAE (i.e., measured as retinol activity equivalents), Vitamin B6, Vitamin B12, Vitamin C, Vitamin D, Vitamin E, Vitamin K, folate, and niacin

We estimated the 2030 Adequacy Ratio (AR)–the ratio of average nutrient availability to RDA for a representative consumer (i.e., adjusted for differences in age and gender requirements) in 2030 –as our metric of sufficient nutrient intake. A value of one indicates adequacy for the average consumer. Our estimates of daily food availability are based on projections of dietary changes over time driven by scenario-specific changes in population, income and other factors [45]. We did not directly estimate the effects of excess energy intake resulting in overweight or obesity. We return to this issue in our discussion of Fig 3A and 3B, which report on the adequacy ratio for the nutrients mentioned above.

## Results

### Total value of production

Table 3 shows the total value of production in 2015 and 2030 for the selected countries as a group. Results for 2015 are shown both as estimated by the parity model (average for 2014–2016), and as modeled by IMPACT. Results for 2030 are modeled by IMPACT (for the reference case, i.e. before the productivity enhancement scenarios are applied). The modeled IMPACT estimates for 2015 broadly match FAO data for this period. (The values for 2015 modeled by IMPACT are broadly similar but not identical to those estimated directly from

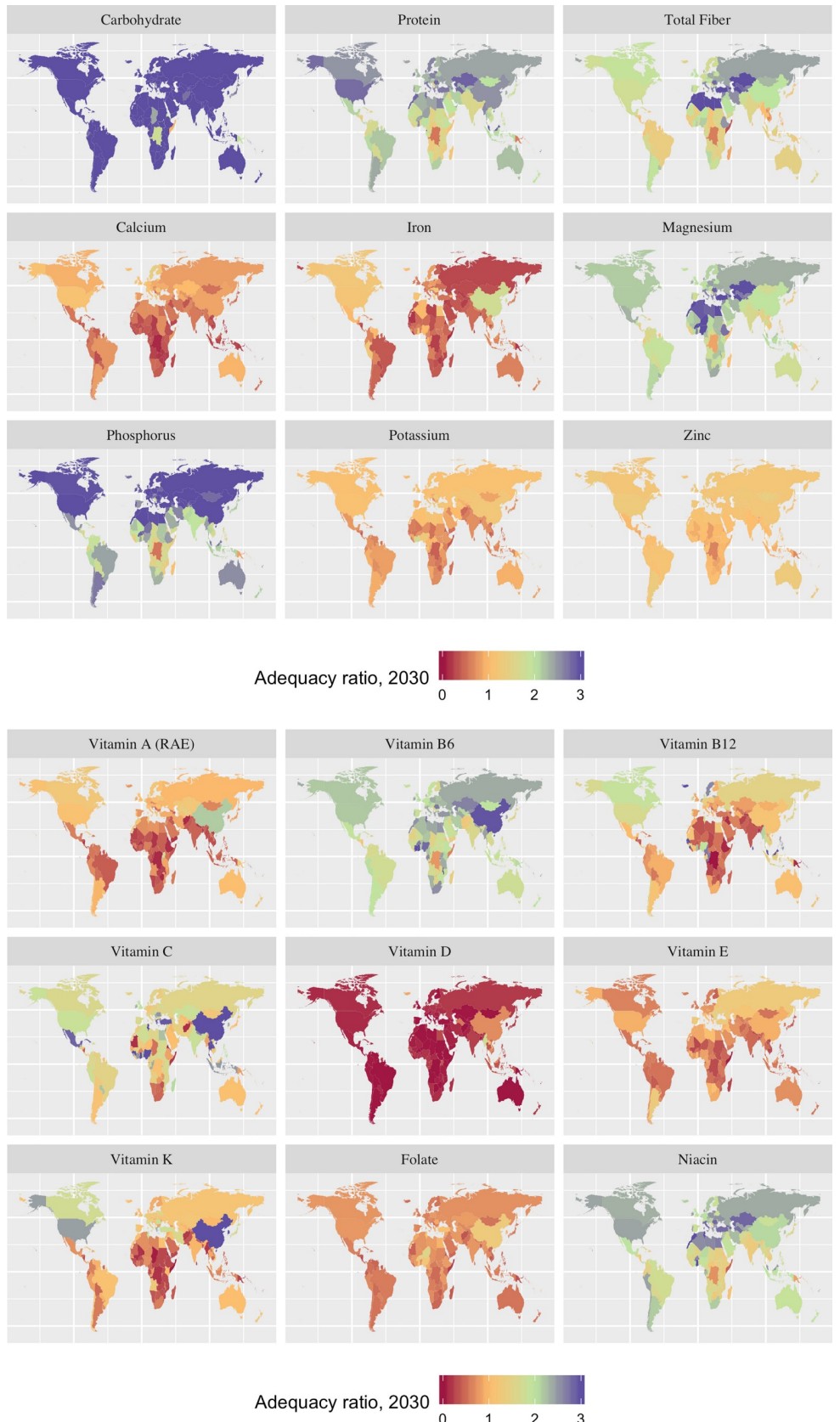

**Fig 3.** Adequacy ratios in the reference case in 2030 for (a) selected macronutrients and minerals, and (b) selected vitamins. Adequacy ratios = 1 where the average daily availability of a nutrient is equal to the RDA for a representative consumer. Source: The authors, based on results from the IMPACT model, using a modeling approach detailed in Nelson et al. [45] with Natural Earth map files (https://www.naturalearthdata.com/) using ggplot2 [47] in R [48].

FAO data because IMPACT first applies an algorithm to reconcile inconsistencies in the FAO data, and because IMPACT models a "trend" value for 2015 whereas observed values reflect fluctuations from year to year, even when averaged over several years.)

Not surprisingly, total values are highest for the major staple crops, especially rice, wheat and maize, reflecting the scale of their production and consumption. Between 2015 and 2030, maize, potato, yams, banana, plantain and cowpea production are projected to grow by 70% or more in value terms, while rice, barley, lentils, other pulses and soybean are projected to grow by 35% or less (see last column of Table 3). Even so, changes in each crop's share of the total value of production over this period are relatively small. The value share of rice, the largest crop of the group in terms of value of production, is expected to fall from 28.8% to 26.0%.

When values are weighted by World Bank poverty measures, the share of some crops declines (e.g. for rice, wheat, potato) and the share of other crops increases (e.g. for cassava, yams, cowpea, and groundnuts), reflecting the latter crops' importance for poorer consumers and countries.

The crops accounting for the largest share of the value of production in 2015 vary by region (**Table 4**). Cassava and yams dominate in Sub-Saharan Africa, followed by maize. In South Asia the largest values are for rice and wheat, followed by potato; in Southeast Asia rice dominates by far, followed by cassava; in WANA-CAC wheat and potato; and in LAC maize and banana, followed by rice. Shares also vary across sub-regions within Sub-Saharan Africa (**Table 5**). Cassava and plantain dominate in Central Africa, followed by groundnut; in Southern Africa maize and cassava have the highest value, followed by banana; and in Western Africa yams and cassava lead, followed by rice. Crop shares are more evenly distributed in Eastern Africa, with maize, pulses (especially beans), banana and millet all representing 10–20% of total value.

Note that applying poverty weights to crop values significantly affects the relative importance of crops across regions but the effect is less within regions. For example, the value share of cassava for the selected countries as a group is estimated to be 7.8% in 2015 based on FAO data, but when weighted by the poverty gap it rises to 23.0%. Within Sub-Saharan Africa, cassava's value share is 19.2%, which rises to 25.9% when weighted by the poverty gap. This reflects the larger differences in poverty rates across regions.

## Changes in economy-wide income (economic surplus)

For the selected countries as a group, projected changes in economy-wide income (economic surplus) between 2015 and 2030 due to productivity enhancement are largest for rice, wheat and maize (reflecting the scale of their production and consumption), followed by yams and banana (**Table 6**). Faster productivity growth in rice, wheat and maize increased economy-wide income in the selected countries in 2030 by 59 billion USD, 27 billion USD and 21 billion USD (about 11 USD, 5 USD and 4 USD per capita) respectively, followed by banana and yams with increases of 9 billion USD each. As was true for the total value of production, for the selected countries as a group, the income results change when weighted by the poverty headcount or poverty gap. Poverty-weighted income shares decline for rice and wheat, for example, reflecting the dominance of richer countries in the production and utilization of those crops, and increase for crops such as sorghum, millet, yams, and groundnut, which are relatively

Table 3. Parity model results: Gross production value from FAOSTAT in 2015, and as modeled by the IMPACT model for 2015 and 2030.

| | PARITY MODEL from FAO Data: 2015 | | | | PARITY MODEL with IMPACT MODEL PROJECTIONS: 2015 | | | | PARITY MODEL with IMPACT MODEL PROJECTIONS: 2030 | | | | IMPACT PARITY |
|---|---|---|---|---|---|---|---|---|---|---|---|---|---|
| Commodity | Gross Production Value (million $) | Value Share (VS) (%) | VS weighted by | | Gross Production Value (million $) | Value Share (VS) (%) | VS weighted by | | Gross Production Value (million $) | Value Share (VS) (%) | VS weighted by | | Ratio of 2030 GPV to 2015 GPV |
| | | | poverty count (%) | poverty gap (%) | | | poverty count (%) | poverty gap (%) | | | poverty count (%) | poverty gap (%) | |
| **Cereal Grains** | | | | | | | | | | | | | |
| Rice | 119,883 | 36.4 | 25.8 | 14.8 | 95,953 | 28.8 | 20.7 | 11.4 | 129,227 | 26.0 | 18.8 | 11.9 | 1.3 |
| Maize | 27,693 | 8.4 | 7.9 | 8.6 | 28,715 | 8.6 | 7.3 | 7.8 | 47,633 | 9.6 | 7.8 | 7.9 | 1.7 |
| Wheat | 32,768 | 10.0 | 6.1 | 2.1 | 55,993 | 16.8 | 9.4 | 3.0 | 80,665 | 16.2 | 9.2 | 2.9 | 1.4 |
| Sorghum | 4,515 | 1.4 | 2.2 | 1.9 | 8,797 | 2.6 | 3.9 | 4.4 | 13,401 | 2.7 | 4.0 | 4.3 | 1.5 |
| Millet | 7,607 | 2.3 | 2.8 | 3.0 | 7,037 | 2.1 | 3.9 | 4.2 | 10,753 | 2.2 | 4.0 | 4.2 | 1.5 |
| Barley | 2,738 | 0.8 | 0.3 | 0.1 | 3,847 | 1.2 | 0.2 | 0.1 | 5,024 | 1.0 | 0.2 | 0.1 | 1.3 |
| **Roots, Tubers & Bananas** | | | | | | | | | | | | | |
| Potato | 19,503 | 5.9 | 4.5 | 3.0 | 20,277 | 6.1 | 4.8 | 3.5 | 34,515 | 7.0 | 5.0 | 3.0 | 1.7 |
| Cassava | 25,639 | 7.8 | 13.1 | 23.0 | 22,355 | 6.7 | 12.2 | 21.0 | 31,682 | 6.4 | 11.6 | 19.7 | 1.4 |
| Yams | 17,096 | 5.2 | 12.1 | 18.3 | 17,925 | 5.4 | 12.7 | 18.8 | 29,991 | 6.0 | 13.6 | 19.5 | 1.7 |
| Sweet potato | 2,126 | 0.6 | 1.2 | 1.7 | 2,149 | 0.6 | 1.2 | 1.7 | 3,333 | 0.7 | 1.2 | 1.7 | 1.6 |
| Banana | 22,400 | 6.8 | 6.5 | 5.7 | 27,562 | 8.3 | 7.3 | 6.5 | 46,410 | 9.3 | 8.1 | 6.9 | 1.7 |
| Plantain | 8,580 | 2.6 | 2.9 | 3.4 | 11,040 | 3.3 | 4.8 | 5.9 | 19,526 | 3.9 | 5.5 | 6.3 | 1.8 |
| **Oilseeds & Pulses** | | | | | | | | | | | | | |
| Pulses, total | 22,576 | 6.9 | 8.0 | 7.5 | 22,975 | 6.9 | 7.8 | 7.2 | 32,924 | 6.6 | 7.5 | 7.1 | 1.4 |
| Beans | 11,622 | 3.5 | 3.7 | 3.9 | 8,772 | 2.6 | 2.6 | 2.6 | 11,986 | 2.4 | 2.3 | 2.5 | 1.4 |
| Chickpea | 5,292 | 1.6 | 1.6 | 0.7 | 5,513 | 1.7 | 1.4 | 0.5 | 8,191 | 1.6 | 1.4 | 0.5 | 1.5 |
| Cowpea | 2,038 | 0.6 | 1.5 | 2.0 | 2,516 | 0.8 | 1.9 | 2.7 | 4,617 | 0.9 | 2.2 | 3.1 | 1.8 |
| Pigeonpea | 2,430 | 0.7 | 0.9 | 0.8 | 2,671 | 0.8 | 0.8 | 0.5 | 4,050 | 0.8 | 0.8 | 0.5 | 1.5 |
| Lentil | 824 | 0.3 | 0.2 | 0.1 | 1,166 | 0.4 | 0.2 | 0.1 | 1,401 | 0.3 | 0.2 | 0.1 | 1.2 |
| other pulses | 369 | 0.1 | 0.1 | 0.1 | 2,337 | 0.7 | 0.8 | 0.7 | 2,679 | 0.5 | 0.6 | 0.5 | 1.1 |
| Groundnuts | 10,826 | 3.3 | 5.1 | 6.0 | 7,223 | 2.2 | 3.3 | 4.2 | 9,988 | 2.0 | 3.1 | 4.2 | 1.4 |
| Soybean | 5,180 | 1.6 | 1.5 | 0.9 | 1,188 | 0.4 | 0.5 | 0.4 | 1,509 | 0.3 | 0.4 | 0.4 | 1.3 |
| **Totals by region** | | | | | | | | | | | | | |
| SSA | 83,992 | 25.5 | 57.0 | 87.9 | 84,513 | 25.4 | 57.9 | 88.6 | 139,735 | 28.1 | 60.6 | 89.6 | |
| LAC | 23,805 | 7.2 | 1.8 | 0.4 | 27,563 | 8.3 | 1.9 | 0.4 | 42,951 | 8.6 | 1.9 | 0.4 | |
| Asia | 194,440 | 59.1 | 40.9 | 11.7 | 180,751 | 54.3 | 39.7 | 10.9 | 256,892 | 51.7 | 37.0 | 10.0 | |
| WANA-CAC | 26,892 | 8.2 | 0.4 | 0.0 | 40,210 | 12.1 | 0.5 | 0.0 | 57,002 | 11.5 | 0.5 | 0.0 | |
| **Total for all crops** | 329,129 | 100.0 | 100.0 | 100.0 | 333,036 | 100.0 | 100.0 | 100.0 | 496,579 | 100.0 | 100.0 | 100.0 | |

Notes: 1. Results from FAO for 2015 are averages for 2014–2016, using global average commodity prices from 2004–06 (i.e., in constant 2005 international dollars). 2. Value weighted by poverty headcount: Value in each country is multiplied by its $1.9/day poverty headcount index (share of population earning less than $1.9/day). 3. Value weighted by poverty gap: Value in each country is multiplied by its poverty headcount index times its poverty gap index (the difference between $1.9 and the mean income of the poor in a country, expressed as a percent of $1.9).

Sources: The authors, based on FAOSTAT (production value), IFPRI (IMPACT projections), PovcalNet (poverty measures, latest available year).

more important in poorer countries. Poverty weighting also increases the share of increased income (i.e., the share of total benefits accruing to poor households) accounted for by Sub-Saharan Africa, while decreasing it in the other regions. Growth in unweighted economy-wide

**Table 4. Parity model results: Gross production value from FAOSTAT in 2015, by region.**

| Commodity | Sub-Saharan Africa | | | | South Asia | | | | Southeast Asia | | | | West Asia, North Africa, and Central Asia | | | | Latin America and Caribbean (excl. Brazil, Southern Cone) | | | |
|---|---|---|---|---|---|---|---|---|---|---|---|---|---|---|---|---|---|---|---|---|
| | Gross Production Value (million $) | Value Share (VS) (%) | VS weighted by poverty count (%) | VS weighted by poverty gap (%) | Gross Production Value (million $) | Value Share (VS) (%) | VS weighted by poverty count (%) | VS weighted by poverty gap (%) | Gross Production Value (million $) | Value Share (VS) (%) | VS weighted by poverty count (%) | VS weighted by poverty gap (%) | Gross Production Value (million $) | Value Share (VS) (%) | VS weighted by poverty count (%) | VS weighted by poverty gap (%) | Gross Production Value (million $) | Value Share (VS) (%) | VS weighted by poverty count (%) | VS weighted by poverty gap (%) |
| **Cereal Grains** | | | | | | | | | | | | | | | | | | | | |
| Rice | 7,045 | 8.4 | 9.7 | 10.7 | 49,295 | 44.7 | 45.4 | 45.7 | 57,827 | 68.6 | 65.0 | 48.9 | 2,687 | 10.0 | 11.6 | 2.3 | 3,028 | 12.7 | 10.5 | 8.5 |
| Maize | 9,299 | 11.1 | 9.9 | 9.2 | 4,708 | 4.3 | 3.9 | 3.8 | 5,820 | 6.9 | 8.1 | 7.0 | 2,553 | 9.5 | 11.9 | 6.9 | 5,313 | 22.3 | 18.2 | 13.1 |
| Wheat | 1,209 | 1.4 | 1.0 | 0.5 | 19,923 | 18.1 | 16.0 | 15.4 | 25 | 0.0 | 0.0 | 0.0 | 10,941 | 40.7 | 31.3 | 17.1 | 670 | 2.8 | 2.0 | 0.8 |
| Sorghum | 2,319 | 2.8 | 2.6 | 1.9 | 2,129 | 1.9 | 2.1 | 2.1 | 44 | 0.1 | 0.1 | 0.0 | 22 | 0.1 | 1.0 | 3.9 | 0 | 0.0 | 0.0 | 0.0 |
| Millet | 4,280 | 5.1 | 4.4 | 3.3 | 811 | 0.7 | 0.8 | 0.8 | 45 | 0.1 | 0.1 | 0.1 | 1,311 | 4.9 | 6.9 | 23.6 | 1,161 | 4.9 | 4.0 | 2.9 |
| Barley | 302 | 0.4 | 0.3 | 0.1 | 308 | 0.3 | 0.2 | 0.2 | 2 | 0.0 | 0.0 | 0.0 | 1,990 | 7.4 | 4.8 | 2.3 | 136 | 0.6 | 0.4 | 0.1 |
| **Roots, Tubers & Bananas** | | | | | | | | | | | | | | | | | | | | |
| Potato | 2,218 | 2.6 | 2.3 | 2.3 | 9,580 | 8.7 | 8.8 | 8.7 | 406 | 0.5 | 0.6 | 0.4 | 5,344 | 19.9 | 21.4 | 22.1 | 1,956 | 8.2 | 7.3 | 4.9 |
| Cassava | 16,133 | 19.2 | 21.8 | 25.9 | 626 | 0.6 | 0.6 | 0.6 | 8,263 | 9.8 | 6.2 | 6.5 | 0 | 0.0 | 0.0 | 0.0 | 616 | 2.6 | 3.1 | 4.7 |
| Yams | 16,689 | 19.9 | 21.0 | 20.7 | 0 | 0.0 | 0.0 | 0.0 | 113 | 0.1 | 0.8 | 5.3 | 0 | 0.0 | 0.0 | 0.0 | 294 | 1.2 | 3.1 | 7.3 |
| Sweet potato | 1,493 | 1.8 | 1.9 | 1.8 | 97 | 0.1 | 0.1 | 0.1 | 389 | 0.5 | 0.9 | 3.3 | 32 | 0.1 | 0.2 | 0.0 | 115 | 0.5 | 1.0 | 2.5 |
| Banana | 4,681 | 5.6 | 5.3 | 5.2 | 7,243 | 6.6 | 7.2 | 7.3 | 4,650 | 5.5 | 9.0 | 21.9 | 600 | 2.2 | 4.6 | 10.0 | 5,226 | 22.0 | 25.0 | 25.3 |
| Plantain | 4,898 | 5.8 | 4.6 | 3.8 | 159 | 0.1 | 0.0 | 0.0 | 1,037 | 1.2 | 1.7 | 1.3 | 0 | 0.0 | 0.0 | 0.0 | 2,486 | 10.4 | 9.9 | 9.7 |
| **Oilseeds & Pulses** | | | | | | | | | | | | | | | | | | | | |
| Pulses, total | 6,780 | 8.1 | 7.9 | 7.4 | 9,070 | 8.2 | 8.8 | 8.9 | 3,992 | 4.7 | 5.3 | 3.7 | 1,108 | 4.1 | 5.4 | 11.6 | 1,627 | 6.8 | 8.7 | 12.6 |
| beans | 3,847 | 4.6 | 4.2 | 4.0 | 2,570 | 2.3 | 2.5 | 2.6 | 3,342 | 4.0 | 4.4 | 3.1 | 426 | 1.6 | 2.0 | 1.5 | 1,438 | 6.0 | 6.9 | 8.6 |
| chickpea | 323 | 0.4 | 0.3 | 0.2 | 4,196 | 3.8 | 4.0 | 4.1 | 275 | 0.3 | 0.4 | 0.3 | 428 | 1.6 | 2.8 | 8.8 | 70 | 0.3 | 0.2 | 0.1 |
| cowpea | 1,975 | 2.4 | 2.6 | 2.3 | 5 | 0.0 | 0.0 | 0.0 | 39 | 0.0 | 0.1 | 0.0 | 3 | 0.0 | 0.0 | 0.0 | 17 | 0.1 | 0.2 | 0.6 |
| pigeonpea | 455 | 0.5 | 0.6 | 0.7 | 1,581 | 1.4 | 1.6 | 1.6 | 322 | 0.4 | 0.4 | 0.3 | 0 | 0.0 | 0.0 | 0.0 | 73 | 0.3 | 1.3 | 3.4 |
| lentil | 63 | 0.1 | 0.1 | 0.0 | 522 | 0.5 | 0.5 | 0.5 | 0 | 0.0 | 0.0 | 0.0 | 234 | 0.9 | 0.5 | 1.1 | 5 | 0.0 | 0.0 | 0.0 |
| other pulses | 117 | 0.1 | 0.1 | 0.1 | 197 | 0.2 | 0.2 | 0.2 | 14 | 0.0 | 0.0 | 0.0 | 17 | 0.1 | 0.1 | 0.2 | 23 | 0.1 | 0.1 | 0.1 |
| Groundnuts | 5,997 | 7.1 | 6.9 | 6.4 | 3,214 | 2.9 | 3.2 | 3.2 | 1,228 | 1.5 | 1.7 | 1.3 | 209 | 0.8 | 0.7 | 0.2 | 176 | 0.7 | 0.9 | 1.2 |
| Soybean | 651 | 0.8 | 0.7 | 0.6 | 3,023 | 2.7 | 3.0 | 3.1 | 412 | 0.5 | 0.5 | 0.4 | 95 | 0.4 | 0.2 | 0.1 | 999 | 4.2 | 5.9 | 6.3 |
| **Totals** | | | | | | | | | | | | | | | | | | | | |
| Cereal Grains | 24,453 | 29.1 | 27.8 | 25.7 | 77,175 | 70.0 | 68.5 | 68.1 | 63,763 | 75.7 | 73.3 | 56.0 | 19,504 | 72.5 | 67.6 | 56.1 | 10,309 | 43.3 | 35.1 | 25.4 |
| Roots, Tubers & Bananas | 46,111 | 54.9 | 56.8 | 59.9 | 17,705 | 16.1 | 16.6 | 16.7 | 14,857 | 17.6 | 19.2 | 38.6 | 5,976 | 22.2 | 26.2 | 32.1 | 10,694 | 44.9 | 49.4 | 54.4 |
| Oilseeds & Pulses | 13,428 | 16.0 | 15.4 | 14.4 | 15,307 | 13.9 | 14.9 | 15.2 | 5,633 | 6.7 | 7.5 | 5.4 | 1,412 | 5.3 | 6.3 | 11.8 | 2,802 | 11.8 | 15.5 | 20.2 |
| **Total for all crops** | 83,992 | 100.0 | 100.0 | 100.0 | 110,187 | 100.0 | 100.0 | 100.0 | 84,252 | 100.0 | 100.0 | 100.0 | 26,892 | 100.0 | 100.0 | 100.0 | 23,805 | 100.0 | 100.0 | 100.0 |

Notes: 1. Results from FAO for 2015 are averages for 2014–2016, using global average commodity prices from 2004–06 (i.e., in constant 2005 international dollars). 2. Value weighted by poverty headcount: Value in each country is multiplied by its $1.9/day poverty headcount index (share of population earning less than $1.9/day). 3. Value weighted by poverty gap: Value in each country is multiplied by its poverty headcount index times its poverty gap index (the difference between $1.9 and the mean income of the poor in a country, expressed as a percent of $1.9).

Sources: The authors, based on FAOSTAT (production value), PovcalNet (poverty measures, latest available year).

**Table 5. Parity model results: Gross production value from FAOSTAT in 2015, by subregion in Sub-Saharan Africa.**

| Commodity | SSA, Central | | | | SSA, Eastern | | | | SSA, Southern | | | | SSA, Western | | | |
|---|---|---|---|---|---|---|---|---|---|---|---|---|---|---|---|---|
| | Gross Production Value (million $) | Value Share (VS) (%) | VS weighted by | | Gross Production Value (million $) | Value Share (VS) (%) | VS weighted by | | Gross Production Value (million $) | Value Share (VS) (%) | VS weighted by | | Gross Production Value (million $) | Value Share (VS) (%) | VS weighted by | |
| | | | poverty count (%) | poverty gap (%) | | | poverty count (%) | poverty gap (%) | | | poverty count (%) | poverty gap (%) | | | poverty count (%) | poverty gap (%) |
| **Cereal Grains** | | | | | | | | | | | | | | | | |
| Rice | 163 | 2.5 | 3.1 | 3.4 | 194 | 1.3 | 1.5 | 1.8 | 1,968 | 11.6 | 15.7 | 20.8 | 4,719 | 10.3 | 10.1 | 9.2 |
| Maize | 479 | 7.4 | 9.0 | 9.9 | 2,105 | 14.5 | 12.2 | 8.7 | 3,863 | 22.7 | 18.4 | 16.4 | 2,852 | 6.2 | 6.4 | 6.4 |
| Wheat | 1 | 0.0 | 0.0 | 0.0 | 857 | 5.9 | 5.7 | 3.4 | 332 | 2.0 | 0.9 | 0.5 | 19 | 0.0 | 0.0 | 0.0 |
| Sorghum | 30 | 0.5 | 0.6 | 0.6 | 440 | 3.0 | 2.3 | 1.2 | 98 | 0.6 | 0.6 | 0.4 | 1,751 | 3.8 | 3.8 | 3.1 |
| Millet | 195 | 3.0 | 3.7 | 4.0 | 1,753 | 12.0 | 9.0 | 4.6 | 234 | 1.4 | 1.3 | 1.1 | 2,098 | 4.6 | 4.9 | 4.8 |
| Barley | 0 | 0.0 | 0.0 | 0.0 | 254 | 1.7 | 1.8 | 1.0 | 48 | 0.3 | 0.1 | 0.0 | 0 | 0.0 | 0.0 | 0.0 |
| **Roots, Tubers & Bananas** | | | | | | | | | | | | | | | | |
| Potato | 71 | 1.1 | 1.3 | 1.5 | 797 | 5.5 | 5.1 | 6.5 | 1,070 | 6.3 | 5.3 | 4.9 | 280 | 0.6 | 0.7 | 0.7 |
| Cassava | 2,334 | 35.9 | 43.8 | 48.1 | 1,005 | 6.9 | 11.1 | 17.3 | 3,287 | 19.3 | 22.3 | 25.2 | 9,507 | 20.7 | 20.4 | 21.2 |
| Yams | 359 | 5.5 | 6.7 | 7.4 | 340 | 2.3 | 2.3 | 1.5 | 4 | 0.0 | 0.0 | 0.0 | 15,985 | 34.8 | 35.9 | 38.4 |
| Sweet potato | 56 | 0.9 | 1.0 | 1.1 | 499 | 3.4 | 4.0 | 4.4 | 543 | 3.2 | 3.3 | 3.1 | 395 | 0.9 | 1.0 | 1.0 |
| Banana | 411 | 6.3 | 2.7 | 0.7 | 1,814 | 12.5 | 16.4 | 24.6 | 2,208 | 13.0 | 11.6 | 9.0 | 247 | 0.5 | 0.4 | 0.3 |
| Plantain | 1,386 | 21.3 | 8.8 | 2.2 | 1,043 | 7.2 | 7.8 | 5.2 | 238 | 1.4 | 1.7 | 1.7 | 2,231 | 4.9 | 3.4 | 2.8 |
| **Oilseeds & Pulses** | | | | | | | | | | | | | | | | |
| Pulses, total | 455 | 7.0 | 8.5 | 9.4 | 2,415 | 16.6 | 16.3 | 17.6 | 1,727 | 10.1 | 10.9 | 10.0 | 2,183 | 4.8 | 5.0 | 4.6 |
| beans | 365 | 5.6 | 6.8 | 7.5 | 1,818 | 12.5 | 13.3 | 15.8 | 1,178 | 6.9 | 6.9 | 5.6 | 486 | 1.1 | 0.9 | 0.8 |
| chickpea | 0 | 0.0 | 0.0 | 0.0 | 241 | 1.7 | 1.7 | 1.0 | 82 | 0.5 | 0.6 | 0.6 | 1 | 0.0 | 0.0 | 0.0 |
| cowpea | 85 | 1.3 | 1.6 | 1.8 | 90 | 0.6 | 0.1 | 0.1 | 106 | 0.6 | 0.7 | 0.7 | 1,693 | 3.7 | 4.1 | 3.8 |
| pigeonpea | 3 | 0.0 | 0.1 | 0.1 | 130 | 0.9 | 0.1 | 0.1 | 322 | 1.9 | 2.4 | 2.8 | 0 | 0.0 | 0.0 | 0.0 |
| lentil | 0 | 0.0 | 0.0 | 0.0 | 62 | 0.4 | 0.4 | 0.3 | 1 | 0.0 | 0.0 | 0.0 | 0 | 0.0 | 0.0 | 0.0 |
| other pulses | 2 | 0.0 | 0.0 | 0.0 | 74 | 0.5 | 0.6 | 0.4 | 39 | 0.2 | 0.3 | 0.3 | 3 | 0.0 | 0.0 | 0.0 |
| Groundnuts | 556 | 8.5 | 10.4 | 11.5 | 972 | 6.7 | 3.9 | 1.7 | 1,061 | 6.2 | 6.6 | 5.9 | 3,408 | 7.4 | 7.4 | 7.0 |
| Soybean | 12 | 0.2 | 0.2 | 0.3 | 77 | 0.5 | 0.6 | 0.5 | 353 | 2.1 | 1.2 | 0.9 | 209 | 0.5 | 0.5 | 0.6 |
| **Totals** | | | | | | | | | | | | | | | | |
| Cereal Grains | 869 | 13.3 | 16.3 | 17.9 | 5,602 | 38.5 | 32.5 | 20.8 | 6,544 | 38.4 | 37.1 | 39.3 | 11,438 | 24.9 | 25.4 | 23.4 |
| Roots, Tubers & Bananas | 4,617 | 70.9 | 64.5 | 61.0 | 5,498 | 37.8 | 46.7 | 59.5 | 7,350 | 43.1 | 44.2 | 43.9 | 28,645 | 62.4 | 61.7 | 64.4 |
| Oilseeds & Pulses | 1,024 | 15.7 | 19.2 | 21.1 | 3,465 | 23.8 | 20.8 | 19.7 | 3,141 | 18.4 | 18.7 | 16.8 | 5800 | 12.6 | 12.9 | 12.2 |
| **Total for all crops** | 6,509 | 100.0 | 100.0 | 100.0 | 14,565 | 100.0 | 100.0 | 100.0 | 17,035 | 100.0 | 100.0 | 100.0 | 45,883 | 100.0 | 100.0 | 100.0 |

Notes: 1. Results from FAO for 2015 are averages for 2014–2016, using global average commodity prices from 2004–06 (i.e., in constant 2005 international dollars). 2. Value weighted by poverty headcount: Value in each country is multiplied by its $1.9/day poverty headcount index (share of population earning less than $1.9/day). 3. Value weighted by poverty gap: Value in each country is multiplied by its poverty headcount index times its poverty gap index (the difference between $1.9 and the mean income of the poor in a country, expressed as a percent of $1.9). 4. SSA subregions are as defined in Table 2.
Sources: The authors, based on FAOSTAT (production value), PovcalNet (poverty measures, latest available year).

income was largest in South Asia, but when weighted by the poverty gap, the largest increase was estimated to have occurred in Sub-Saharan Africa.

Rice, maize, sorghum, yams and millet represent the largest shares of economic surplus in 2030 in Sub-Saharan Africa (Table 7); rice and wheat in South Asia and WANA-CAC; rice in Southeast Asia; and maize followed by rice and wheat in LAC. Economic surplus is highest for maize, rice and cassava in Central Africa (Table 8); for maize, sorghum, millet, wheat and plantain in Eastern Africa; for maize in Southern Africa; and for rice, yams and sorghum in

**Table 6. Economic surplus model results: Change in economy-wide income in 2030 from faster productivity growth, as modeled by the IMPACT model.**

| Commodity/ Scenario | ECONOMIC SURPLUS MODEL | | | |
|---|---|---|---|---|
| | Economic Surplus (ES) (million $) | ES share (ESS) (%) | ESS weighted by | |
| | | | poverty count (%) | poverty gap (%) |
| **Cereal Grains** | | | | |
| Rice | 59,256 | 35.6 | 29.3 | 23.4 |
| Maize | 20,722 | 12.4 | 10.8 | 10.7 |
| Wheat | 26,560 | 15.9 | 12.8 | 5.6 |
| Sorghum | 8,011 | 4.8 | 8.7 | 13.8 |
| Millet | 6,219 | 3.7 | 7.0 | 11.1 |
| Barley | 2,802 | 1.7 | 1.6 | 0.8 |
| **Roots, Tubers & Bananas** | | | | |
| Potato | 4,607 | 2.8 | 2.1 | 1.0 |
| Cassava | 4,310 | 2.6 | 3.3 | 4.9 |
| Yams | 9,104 | 5.5 | 8.7 | 13.6 |
| Sweet potato | 708 | 0.4 | 0.4 | 0.5 |
| Banana | 9,342 | 5.6 | 4.8 | 2.1 |
| Plantain | 3,000 | 1.8 | 1.9 | 2.5 |
| **Oilseeds & Pulses** | | | | |
| Pulses, total | 7,464 | 4.5 | 4.3 | 3.2 |
| Beans | 1,547 | 0.9 | 0.8 | 0.5 |
| Chickpea | 2,681 | 1.6 | 1.4 | 0.6 |
| Cowpea | 1,187 | 0.7 | 1.0 | 1.6 |
| Pigeonpea | 1,137 | 0.7 | 0.6 | 0.3 |
| Lentil | 413 | 0.2 | 0.2 | 0.1 |
| other pulses | 499 | 0.3 | 0.2 | 0.2 |
| Groundnuts | 4,257 | 2.6 | 4.0 | 6.5 |
| Soybean | 181 | 0.1 | 0.2 | 0.3 |
| **Totals by region** | | | | |
| SSA | 35,930 | 21.6 | 46.5 | 82.0 |
| LAC | 2,961 | 1.8 | 0.3 | 0.1 |
| Asia | 113,959 | 68.4 | 52.9 | 17.9 |
| WANA-CAC | 13,692 | 8.2 | 0.3 | 0.0 |
| **Total for all crops** | 166,541 | 100.0 | 100.0 | 100.0 |

Notes: 1. ES weighted by poverty headcount: ES in each country is multiplied by its $1.9/day poverty headcount index (share of population earning less than $1.9/day). 2. ES weighted by poverty gap: ES in each country is multiplied by its poverty headcount index and its poverty gap index (the difference between $1.9 and the mean income of the poor in a country, expressed as a percent of $1.9). 3. Totals are indicative because the crop scenarios were run separately for each crop, i.e. ES for crop i is estimated separately for each crop scenario i. 4. Total ES is the sum of ES for all crops estimated separately, and ESS is the share of total ES that is accounted for by each crop or region.

Sources: The authors, based on IFPRI (economic surplus projections), PovcalNet (poverty measures, latest available year).

**Table 7. Economic surplus model results: Change in economy-wide income in 2030 from faster productivity growth, as modeled by the IMPACT model, by region.**

| Commodity/ Scenario | Sub-Saharan Africa | | | | South Asia | | | | Southeast Asia | | | | West Asia, North Africa, and Central Asia | | | | Latin America and Caribbean (excl. Brazil, Southern Cone) | | | |
|---|---|---|---|---|---|---|---|---|---|---|---|---|---|---|---|---|---|---|---|---|
| | Economic Surplus (ES) (million $) | ES share (ESS) (%) | ESS weighted by poverty count (%) | ESS weighted by poverty gap (%) | Economic Surplus (ES) (million $) | ES share (ESS) (%) | ESS weighted by poverty count (%) | ESS weighted by poverty gap (%) | Economic Surplus (ES) (million $) | ES share (ESS) (%) | ESS weighted by poverty count (%) | ESS weighted by poverty gap (%) | Economic Surplus (ES) (million $) | ES share (ESS) (%) | ESS weighted by poverty count (%) | ESS weighted by poverty gap (%) | Economic Surplus (ES) (million $) | ES share (ESS) (%) | ESS weighted by poverty count (%) | ESS weighted by poverty gap (%) |
| **Cereal Grains** | | | | | | | | | | | | | | | | | | | | |
| Rice | 7,051 | 19.6 | 20.6 | 21.0 | 32,283 | 36.2 | 35.1 | 34.0 | 16,132 | 65.4 | 64.7 | 58.6 | 3,400 | 24.8 | 24.2 | 15.1 | 389 | 13.1 | 11.4 | 8.6 |
| Maize | 5,420 | 15.1 | 12.2 | 10.9 | 7,874 | 8.8 | 8.9 | 9.1 | 4,044 | 16.4 | 15.6 | 17.5 | 1,911 | 14.0 | 19.1 | 25.3 | 1,474 | 49.8 | 53.5 | 63.2 |
| Wheat | 1,023 | 2.8 | 2.1 | 1.6 | 20,533 | 23.0 | 23.8 | 24.4 | 207 | 0.8 | 0.6 | 0.6 | 4,465 | 32.6 | 19.6 | 6.1 | 331 | 11.2 | 9.1 | 3.9 |
| Sorghum | 5,412 | 15.1 | 16.0 | 16.3 | 2,086 | 2.3 | 2.4 | 2.5 | 163 | 0.7 | 0.0 | 0.0 | 324 | 2.4 | 5.3 | 8.8 | 26 | 0.9 | 0.7 | 0.3 |
| Millet | 4,383 | 12.2 | 13.0 | 13.1 | 1,665 | 1.9 | 1.9 | 2.0 | 43 | 0.2 | 0.1 | 0.0 | 106 | 0.8 | 6.0 | 20.4 | 22 | 0.7 | 0.6 | 0.2 |
| Barley | 157 | 0.4 | 0.3 | 0.3 | 2,492 | 2.8 | 2.9 | 3.0 | 52 | 0.2 | 0.2 | 0.2 | 95 | 0.7 | 0.9 | 1.1 | 6 | 0.2 | 0.2 | 0.2 |
| **Roots, Tubers & Bananas** | | | | | | | | | | | | | | | | | | | | |
| Potato | 231 | 0.6 | 0.5 | 0.5 | 3,465 | 3.9 | 3.6 | 3.6 | 208 | 0.8 | 0.8 | 0.7 | 626 | 4.6 | 4.1 | 4.7 | 78 | 2.6 | 2.4 | 2.0 |
| Cassava | 1,720 | 4.8 | 5.2 | 5.7 | 1,431 | 1.6 | 1.4 | 1.4 | 826 | 3.3 | 3.4 | 4.4 | 310 | 2.3 | 2.1 | 2.8 | 23 | 0.8 | 0.7 | 1.3 |
| Yams | 4,961 | 13.8 | 15.2 | 15.9 | 2,555 | 2.9 | 2.8 | 2.9 | 935 | 3.8 | 5.4 | 8.3 | 586 | 4.3 | 4.0 | 5.3 | 68 | 2.3 | 2.8 | 4.3 |
| Sweet potato | 244 | 0.7 | 0.6 | 0.6 | 269 | 0.3 | 0.3 | 0.3 | 126 | 0.5 | 0.4 | 0.4 | 56 | 0.4 | 0.4 | 0.5 | 13 | 0.4 | 0.4 | 0.4 |
| Banana | 465 | 1.3 | 0.9 | 0.7 | 7,267 | 8.1 | 8.6 | 8.7 | 956 | 3.9 | 4.7 | 5.1 | 565 | 4.1 | 5.9 | 4.3 | 89 | 3.0 | 2.9 | 2.3 |
| Plantain | 1,252 | 3.5 | 3.1 | 2.9 | 961 | 1.1 | 0.9 | 0.9 | 312 | 1.3 | 1.0 | 1.0 | 221 | 1.6 | 1.5 | 2.0 | 253 | 8.6 | 9.9 | 9.7 |
| **Oilseeds & Pulses** | | | | | | | | | | | | | | | | | | | | |
| Pulses, total | 937 | 2.6 | 2.6 | 2.6 | 5,426 | 6.1 | 6.2 | 6.2 | 402 | 1.6 | 1.7 | 1.7 | 532 | 3.9 | 3.9 | 2.4 | 166 | 5.6 | 4.8 | 3.1 |
| beans | 169 | 0.5 | 0.4 | 0.3 | 1,035 | 1.2 | 1.2 | 1.2 | 134 | 0.5 | 0.5 | 0.5 | 117 | 0.9 | 0.7 | 0.6 | 93 | 3.1 | 2.7 | 1.5 |
| chickpea | 63 | 0.2 | 0.1 | 0.1 | 2,379 | 2.7 | 2.7 | 2.7 | 77 | 0.3 | 0.2 | 0.2 | 139 | 1.0 | 0.6 | 0.6 | 22 | 0.7 | 0.6 | 0.3 |
| cowpea | 592 | 1.6 | 1.8 | 1.9 | 316 | 0.4 | 0.3 | 0.3 | 112 | 0.5 | 0.6 | 0.7 | 137 | 1.0 | 1.8 | 0.5 | 31 | 1.0 | 1.0 | 0.9 |
| pigeonpea | 32 | 0.1 | 0.1 | 0.1 | 1,045 | 1.2 | 1.2 | 1.3 | 28 | 0.1 | 0.2 | 0.2 | 26 | 0.2 | 0.2 | 0.2 | 5 | 0.2 | 0.1 | 0.1 |
| lentil | 15 | 0.0 | 0.0 | 0.0 | 332 | 0.4 | 0.4 | 0.4 | 14 | 0.1 | 0.0 | 0.0 | 49 | 0.4 | 0.2 | 0.1 | 4 | 0.1 | 0.1 | 0.1 |
| other pulses | 66 | 0.2 | 0.2 | 0.1 | 319 | 0.4 | 0.3 | 0.3 | 38 | 0.2 | 0.1 | 0.1 | 64 | 0.5 | 0.5 | 0.4 | 11 | 0.4 | 0.4 | 0.3 |
| Groundnuts | 2,527 | 7.0 | 7.3 | 7.7 | 965 | 1.1 | 1.1 | 1.1 | 256 | 1.0 | 1.3 | 1.2 | 486 | 3.6 | 2.8 | 0.7 | 23 | 0.8 | 0.6 | 0.3 |
| Soybean | 147 | 0.4 | 0.3 | 0.3 | 15 | 0.0 | 0.0 | 0.0 | 10 | 0.0 | 0.1 | 0.2 | 9 | 0.1 | 0.2 | 0.4 | 1 | 0.0 | 0.0 | 0.1 |
| **Totals** | | | | | | | | | | | | | | | | | | | | |
| Cereal Grains | 23,446 | 65.3 | 64.2 | 63.2 | 66,933 | 75.0 | 75.1 | 74.9 | 20,641 | 83.7 | 81.3 | 77.1 | 10,301 | 75.2 | 75.0 | 76.9 | 2,247 | 75.9 | 75.5 | 76.5 |
| Roots, Tubers & Bananas | 8,872 | 24.7 | 25.6 | 26.2 | 15,949 | 17.9 | 17.7 | 17.7 | 3,363 | 13.6 | 15.7 | 19.9 | 2,363 | 17.3 | 18.1 | 19.6 | 524 | 17.7 | 19.1 | 19.9 |
| Oilseeds & Pulses | 3,611 | 10.1 | 10.2 | 10.6 | 6,406 | 7.2 | 7.3 | 7.4 | 667 | 2.7 | 3.1 | 3.1 | 1,027 | 7.5 | 6.9 | 3.5 | 189 | 6.4 | 5.5 | 3.6 |
| Total for all crops | 35,930 | 100.0 | 100.0 | 100.0 | 89,288 | 100.0 | 100.0 | 100.0 | 24,671 | 100.0 | 100.0 | 100.0 | 13,692 | 100.0 | 100.0 | 100.0 | 2,961 | 100.0 | 100.0 | 100.0 |

Notes: 1. ES weighted by poverty headcount: ES in each country is multiplied by its $1.9/day poverty headcount index (share of population earning less than $1.9/day). 2. ES (weighted by poverty gap: ES in each country is multiplied by its poverty headcount index and its poverty gap index (the difference between $1.9 and the mean income of the poor in a country, expressed as a percent of $1.9). 3. Totals are indicative because the crop scenarios were run separately for each crop, i.e. ES for crop i is estimated separately for each crop scenario i. Total ES is the sum of ES for all crops estimated separately, and ESS is the share of total ES that is accounted for by each crop or region.

Sources: The authors, based on IFPRI (economic surplus projections), PovcalNet (poverty measures, latest available year).

**Table 8. Economic surplus model results: Change in economy-wide income in 2030 from faster productivity growth, as modeled by the IMPACT model, by subregion in Sub-Saharan Africa.**

| Commodity/ Scenario | SSA, Central | | | | SSA, Eastern | | | | SSA, Southern | | | | SSA, Western | | | |
|---|---|---|---|---|---|---|---|---|---|---|---|---|---|---|---|---|
| | Economic Surplus (ES) (million $) | ES share (ESS) (%) | ESS weighted by poverty count (%) | ESS weighted by poverty gap (%) | Economic Surplus (ES) (million $) | ES share (ESS) (%) | ESS weighted by poverty count (%) | ESS weighted by poverty gap (%) | Economic Surplus (ES) (million $) | ES share (ESS) (%) | ESS weighted by poverty count (%) | ESS weighted by poverty gap (%) | Economic Surplus (ES) (million $) | ES share (ESS) (%) | ESS weighted by poverty count (%) | ESS weighted by poverty gap (%) |
| **Cereal Grains** | | | | | | | | | | | | | | | | |
| Rice | 362 | 15.4 | 14.2 | 13.1 | 210 | 5.4 | 5.1 | 5.7 | 220 | 11.2 | 16.2 | 22.2 | 6,258 | 22.6 | 22.2 | 22.0 |
| Maize | 543 | 23.0 | 21.1 | 17.0 | 1,092 | 28.1 | 26.4 | 30.3 | 1,043 | 53.2 | 48.6 | 43.5 | 2,742 | 9.9 | 9.0 | 8.4 |
| Wheat | 25 | 1.1 | 1.1 | 1.1 | 435 | 11.2 | 13.0 | 11.6 | 219 | 11.1 | 6.5 | 2.5 | 344 | 1.2 | 1.3 | 1.3 |
| Sorghum | 127 | 5.4 | 4.8 | 3.2 | 510 | 13.1 | 12.1 | 9.1 | 25 | 1.3 | 1.0 | 0.8 | 4,750 | 17.1 | 17.7 | 18.1 |
| Millet | 122 | 5.2 | 5.3 | 4.4 | 444 | 11.4 | 11.9 | 9.9 | 28 | 1.4 | 1.1 | 0.8 | 3,790 | 13.7 | 14.1 | 14.4 |
| Barley | 36 | 1.5 | 1.5 | 1.4 | 84 | 2.2 | 2.0 | 2.3 | 21 | 1.1 | 2.0 | 2.8 | 15 | 0.1 | 0.1 | 0.0 |
| **Roots, Tubers & Bananas** | | | | | | | | | | | | | | | | |
| Potato | 39 | 1.6 | 1.5 | 1.5 | 52 | 1.3 | 1.2 | 1.4 | 51 | 2.6 | 2.3 | 2.0 | 89 | 0.3 | 0.3 | 0.3 |
| Cassava | 313 | 13.3 | 16.0 | 21.0 | 84 | 2.2 | 2.6 | 2.9 | 72 | 3.7 | 5.4 | 6.9 | 1,251 | 4.5 | 4.6 | 4.7 |
| Yams | 42 | 1.8 | 1.5 | 1.6 | 64 | 1.6 | 1.6 | 1.3 | 26 | 1.3 | 1.0 | 0.7 | 4,829 | 17.4 | 17.8 | 18.1 |
| Sweet potato | 38 | 1.6 | 1.4 | 1.3 | 39 | 1.0 | 0.9 | 1.1 | 17 | 0.9 | 1.2 | 1.6 | 149 | 0.5 | 0.5 | 0.4 |
| Banana | 200 | 8.5 | 7.4 | 5.5 | 73 | 1.9 | 1.8 | 2.5 | 47 | 2.4 | 2.3 | 2.2 | 144 | 0.5 | 0.4 | 0.3 |
| Plantain | 143 | 6.1 | 6.2 | 7.2 | 399 | 10.3 | 12.9 | 14.6 | 48 | 2.4 | 3.2 | 3.9 | 662 | 2.4 | 2.2 | 2.1 |
| **Oilseeds & Pulses** | | | | | | | | | | | | | | | | |
| Pulses, total | 94 | 4.0 | 3.8 | 3.6 | 132 | 3.4 | 3.5 | 3.6 | 42 | 2.2 | 2.5 | 2.8 | 668 | 2.4 | 2.4 | 2.4 |
| beans | 55 | 2.3 | 2.1 | 1.8 | 45 | 1.2 | 1.1 | 1.3 | 20 | 1.0 | 1.2 | 1.3 | 49 | 0.2 | 0.2 | 0.1 |
| chickpea | 7 | 0.3 | 0.3 | 0.2 | 33 | 0.9 | 1.1 | 1.0 | 5 | 0.3 | 0.3 | 0.3 | 18 | 0.1 | 0.1 | 0.0 |
| cowpea | 17 | 0.7 | 0.8 | 0.9 | 12 | 0.3 | 0.2 | 0.2 | 6 | 0.3 | 0.3 | 0.3 | 558 | 2.0 | 2.1 | 2.1 |
| pigeonpea | 6 | 0.3 | 0.3 | 0.3 | 11 | 0.3 | 0.2 | 0.2 | 3 | 0.2 | 0.3 | 0.3 | 11 | 0.0 | 0.0 | 0.0 |
| lentil | 1 | 0.0 | 0.0 | 0.0 | 8 | 0.2 | 0.3 | 0.3 | 1 | 0.0 | 0.0 | 0.0 | 5 | 0.0 | 0.0 | 0.0 |
| other pulses | 9 | 0.4 | 0.4 | 0.4 | 22 | 0.6 | 0.6 | 0.6 | 7 | 0.3 | 0.4 | 0.5 | 28 | 0.1 | 0.1 | 0.1 |
| Groundnuts | 264 | 11.2 | 13.9 | 18.1 | 249 | 6.4 | 4.7 | 3.1 | 75 | 3.8 | 4.7 | 5.3 | 1,939 | 7.0 | 7.1 | 7.2 |
| Soybean | 11 | 0.4 | 0.3 | 0.3 | 11 | 0.3 | 0.3 | 0.6 | 27 | 1.4 | 2.0 | 2.2 | 98 | 0.4 | 0.3 | 0.2 |
| **Totals** | | | | | | | | | | | | | | | | |
| Cereal Grains | 1,215 | 51.5 | 48.0 | 40.1 | 2,775 | 71.5 | 70.5 | 68.9 | 1,555 | 79.3 | 75.5 | 72.5 | 17,901 | 64.5 | 64.4 | 64.2 |
| Roots, Tubers & Bananas | 775 | 32.8 | 33.9 | 38.1 | 711 | 18.3 | 21.0 | 23.8 | 261 | 13.3 | 15.4 | 17.2 | 7,126 | 25.7 | 25.7 | 25.9 |
| Oilseeds & Pulses | 368 | 15.6 | 18.1 | 21.9 | 392 | 10.1 | 8.5 | 7.3 | 144 | 7.4 | 9.1 | 10.2 | 2,706 | 9.8 | 9.8 | 9.9 |
| **Total for all crops** | 2,359 | 100.0 | 100.0 | 100.0 | 3,879 | 100.0 | 100.0 | 100.0 | 1,960 | 100.0 | 100.0 | 100.0 | 27,732 | 100.00 | 100.00 | 100.00 |

Notes: 1. ES weighted by poverty headcount: ES in each country is multiplied by its $1.9/day poverty headcount index (share of population earning less than $1.9/day). 2. ES weighted by poverty gap: ES in each country is multiplied by its poverty headcount index and its poverty gap index (the difference between $1.9 and the mean income of the poor in a country, expressed as a percent of $1.9). 3. SSA subregions are as defined in Table 2. 5. Totals are indicative because the crop scenarios were run separately for each crop, i.e. ES for crop i is estimated separately for each crop scenario i. 4. Total ES is the sum of ES for all crops estimated separately, and ESS is the share of total ES that is accounted for by each crop or region.

Sources: The authors, based on IFPRI (economic surplus projections), PovcalNet (poverty measures, latest available year).

Western Africa. Poverty weighting makes less difference in the results within regions and sub-regions, as progressively smaller country groupings become more homogeneous.

Faster productivity growth generates economic surplus shares that are generally higher than the parity model's shares of production value for cereals, broadly similar for oilseeds and pulses, and lower for roots, tubers and bananas. Economic surplus shares for the 20 crops in total are higher than parity model shares in Asia, and lower in the other regions. These patterns likely reflect the relative roles of crops in value-added food systems, and the relative importance of cereals in Asia and parts of Africa. Cereal grains are easily stored and traded and widely used by animal feed, food manufacturing and biofuel industries, and thus may have larger multiplier effects in the general economy.

## Hunger and nutrient indicators

Malnutrition today includes substantial populations that suffer from risk of insufficient intake of both energy and micronutrient-rich foods side by side with populations that overconsume energy-rich foods resulting in overweight and obesity [46].

Table 9 presents impacts of the productivity scenarios on the number of undernourished children (suffering from energy intake deficiencies) and the population at risk of hunger in the selected countries in 2030. Improvements (i.e. reductions) are greatest for rice and wheat, which is not surprising since these two measures are based on availability of dietary energy. The plantain, cassava, sorghum, maize and millet scenarios also reduce the population at risk of hunger by a million or more, with roughly proportionate reductions in child undernourishment.

Fig 3 illustrates adequacy ratios in 2030 for a variety of key nutrients (as distinct from total caloric intake) in the reference case without faster productivity growth. In Fig 3(A), almost every country has an adequacy ratio of 3 or greater for carbohydrates, i.e. at least three times the RDA. (Note that because there are three primary sources of caloric intake in a diet–carbohydrates, fats, and protein–it is possible for an individual to consume amounts of carbohydrates well above RDA levels and still have a shortfall in intake of total calories or other essential nutrients.) Protein has adequacy ratios of 3 or above in many countries in the northern hemisphere and above 1 in almost all countries in the world. Calcium, iron, potassium and zinc stand out for global deficiencies. In Fig 3(B), vitamins A, B12, D, E and K, and folate all have widespread deficiencies. Variation is large across nutrients and between countries, highlighting the need for country-specific interventions.

Investments to increase productivity of a particular crop will increase the aggregate availability of the nutrients it contains. Table 10 reports changes to the adequacy ratios due to faster crop productivity growth for 3 macronutrients and 9 micronutrients that are deficient in many countries. Even for the largest crops we see relatively small changes in the adequacy ratios for any of these 12 micronutrients as dietary sources of any nutrient are varied. However, for regions that rely heavily on one staple, the carbohydrate adequacy ratio sees relatively large increases (e.g., rice in Asia and cassava in Sub-Saharan Africa). Crops that have high content of a particular micronutrient can see a substantial increase in its adequacy ratio even with a relatively small share of overall contribution to the diet (e.g., folate from millet, Vitamin E from groundnuts, Vitamin C from cassava and Vitamin A from sweet potato, all in Sub-Saharan Africa). Note that regional aggregation hides the importance of some crops for a specific nutrient in a country. For specific nutrients, we note that:

- None of the yield increases change the zinc adequacy ratios by more than a very small amount because these crops have relatively small zinc content.

**Table 9. Change in undernourished children and population at risk of hunger in 2030 from faster productivity growth.**

| Commodity/ Scenario | Change from Reference Scenario in 2030 | | | | | |
| | Undernourished Children | | | Population at Risk of Hunger | | |
| | (% change) | ('000s) | (% of total) | (% change) | ('000s) | (% of total) |
|---|---|---|---|---|---|---|
| **Cereal Grains** | | | | | | |
| Rice | -0.29 | -360.6 | 35.97 | -2.06 | -10,602.0 | 38.62 |
| Maize | -0.04 | -55.5 | 5.53 | -0.29 | -1,490.7 | 5.43 |
| Wheat | -0.16 | -205.8 | 20.53 | -1.15 | -5,903.8 | 21.51 |
| Sorghum | -0.05 | -61.0 | 6.08 | -0.31 | -1,580.2 | 5.76 |
| Millet | -0.05 | -64.3 | 6.42 | -0.26 | -1,315.7 | 4.79 |
| Barley | 0.00 | -2.0 | 0.20 | -0.01 | -76.1 | 0.28 |
| **Roots, Tubers & Bananas** | | | | | | |
| Potato | -0.01 | -9.5 | 0.95 | -0.06 | -315.0 | 1.15 |
| Cassava | -0.06 | -72.0 | 7.18 | -0.34 | -1,763.8 | 6.43 |
| Yams | -0.03 | -43.4 | 4.33 | -0.09 | -481.6 | 1.75 |
| Sweet potato | -0.01 | -9.0 | 0.90 | -0.06 | -313.2 | 1.14 |
| Banana | -0.01 | -15.7 | 1.56 | -0.09 | -470.4 | 1.71 |
| Plantain | -0.05 | -62.1 | 6.20 | -0.39 | -2,018.6 | 7.35 |
| **Oilseeds & Pulses** | | | | | | |
| Pulses, total | -0.02 | -27.8 | 2.77 | -0.15 | -765.6 | 2.79 |
| beans | -0.01 | -10.6 | 1.05 | -0.08 | -404.5 | 1.47 |
| chickpea | 0.00 | -4.7 | 0.47 | -0.04 | -223.7 | 0.81 |
| cowpea | -0.01 | -6.7 | 0.67 | 0.01 | 46.0 | -0.17 |
| pigeonpea | -0.01 | -9.4 | 0.94 | -0.05 | -242.9 | 0.88 |
| lentil | 0.00 | 4.7 | -0.47 | 0.02 | 115.3 | -0.42 |
| other pulses | 0.00 | -1.1 | 0.11 | -0.01 | -55.8 | 0.20 |
| Groundnuts | -0.01 | -10.2 | 1.02 | -0.05 | -256.1 | 0.93 |
| Soybean | 0.00 | -3.5 | 0.35 | -0.02 | -98.5 | 0.36 |
| **Total for all crops** | | -1,002.5 | 100.00 | | -27,451.5 | 100.00 |

Note: For ease of comparison, the "% of total" scales the relative size of the change in the number of undernourished children or population at risk of hunger associated with each crop so that they sum to 100. Totals are indicative because the crop scenarios were run separately.

Source: The authors, based on results from the IMPACT model.

- In Asia, rice and wheat yield growth improves the adequacy ratios for many nutrients because they make up a large share of total consumption. None of the other yield increases contribute much.

- In Latin American and the Caribbean, wheat yield growth improves adequacy ratios for many nutrients because of its importance as a staple. Maize increases benefit a few adequacy ratios. Rice increases have very little impact on any adequacy ratios.

- In Sub-Saharan Africa, cassava yield growth improves adequacy ratios for many nutrients. Cowpeas, millet, plantain, sorghum, wheat and yams also make improvements in some adequacy ratios. Rice yield increases have little effect on SSA adequacy ratios.

**Table 10. Change in selected nutrient adequacy ratios in 2030 from faster productivity growth.**

| Crop | Region | Carbo-hydrate | Protein | Total Fiber | Iron | Phos-phorus | Potas-sium | Zinc | Vitamin A (RAE) | Vitamin B6 | Vitamin C | Vitamin E | Folate |
|------|--------|------|------|------|------|------|------|------|------|------|------|------|------|
| | | | | | | Percentage change relative to the reference scenario in 2030 | | | | | | | |
| **Cereal Grains** | | | | | | | | | | | | | |
| Maize | LAC | 0.09 | 0.07 | 0.11 | 0.12 | 0.09 | 0.06 | -0.01 | 0.03 | 0.08 | 0.00 | 0.04 | 0.05 |
| Maize | SSA | 0.09 | 0.09 | 0.10 | 0.12 | 0.10 | 0.05 | 0.00 | 0.03 | 0.10 | 0.01 | 0.05 | 0.04 |
| Millet | SSA | 0.12 | 0.13 | 0.15 | 0.13 | 0.15 | 0.05 | 0.00 | -0.01 | 0.11 | 0.01 | 0.02 | 0.16 |
| Rice | EAP | 0.73 | 0.37 | 0.27 | 0.24 | 0.36 | 0.21 | 0.07 | 0.08 | 0.33 | 0.11 | 0.15 | 0.21 |
| Rice | LAC | 0.19 | 0.10 | 0.05 | 0.06 | 0.09 | 0.05 | 0.02 | 0.03 | 0.09 | 0.03 | 0.03 | 0.05 |
| Rice | SAS | 0.48 | 0.31 | 0.20 | 0.15 | 0.26 | 0.23 | 0.06 | 0.18 | 0.33 | 0.21 | 0.15 | 0.18 |
| Rice | SSA | 0.26 | 0.19 | 0.09 | 0.11 | 0.16 | 0.08 | 0.05 | 0.04 | 0.13 | 0.06 | 0.07 | 0.07 |
| Wheat | SAS | 0.33 | 0.37 | 0.59 | 0.46 | 0.40 | 0.30 | 0.09 | 0.11 | 0.33 | 0.09 | 0.33 | 0.27 |
| Wheat | SSA | 0.13 | 0.15 | 0.21 | 0.16 | 0.16 | 0.09 | 0.05 | 0.01 | 0.10 | 0.01 | 0.11 | 0.10 |
| **Roots, Tubers & Bananas** | | | | | | | | | | | | | |
| Cassava | SSA | 0.38 | 0.11 | 0.21 | 0.15 | 0.10 | 0.31 | 0.07 | 0.01 | 0.17 | 0.58 | 0.11 | 0.31 |
| Plantain | SSA | 0.11 | 0.02 | 0.08 | 0.13 | 0.03 | 0.16 | 0.01 | 0.22 | 0.17 | 0.14 | 0.01 | 0.03 |
| Sweet potato | SSA | 0.01 | 0.00 | 0.02 | 0.01 | 0.01 | 0.02 | 0.00 | 0.50 | 0.02 | 0.00 | 0.01 | 0.01 |
| Yam | SSA | 0.12 | 0.05 | 0.19 | 0.14 | 0.08 | 0.34 | 0.03 | 0.06 | 0.22 | 0.19 | 0.09 | 0.13 |
| **Oilseeds & Pulses** | | | | | | | | | | | | | |
| Groundnuts | SSA | 0.02 | 0.06 | 0.04 | 0.05 | 0.05 | 0.03 | 0.01 | 0.02 | 0.03 | 0.01 | 0.15 | 0.08 |

Note: EAP = East Asia and Pacific, LAC = Latin American and Caribbean, SAS = South Asia, SSA = Sub-Saharan Africa.

Source: The authors, based on results from the IMPACT model, using a modeling approach detailed in Nelson et al. [45] with Natural Earth map files (https://www.naturalearthdata.com/) using ggplot2 [47] in R [48].

Beyond contributions to adequacy of nutrient intake, agricultural productivity investments can also affect dietary diversity. Several measures of diversity are available. For this report, we use the non-staple share of energy intake, which is widely used in the nutrition literature [49]. Fig 4 clearly shows the heavy dependence on starchy staples for dietary energy in most of Africa and parts of Central Asia. Impacts of the productivity scenarios on this indicator are generally small. Most of the crops considered in this analysis are in the staple category, so increasing their productivity (and reducing their prices) relative to the other crops generally increases their consumption and decreases the non-staple share of energy intake. The only yield increases that raise the non-staple share more than 0.01 percent are for groundnuts in SSA (0.08 percent) and bananas in Asia and SSA (0.03 and 0.04 respectively). (In this analysis bananas are considered a non-staple, while plantains are considered a staple.)

## Comparing results across different indicators

Fig 5 and Table 11 summarize the different metrics explored in this analysis and help illustrate their implications for R&D allocation. For each of the metrics presented, a "parity rule" would suggest that the share of that metric represented by a particular crop could help inform an efficient R&D allocation. Importantly, the metrics help illustrate how different system goals might influence R&D allocation decisions. The crop value and economic surplus value shares give greater emphasis to total income growth; economic surplus weighted by the poverty indices gives greater emphasis to poverty reduction; while the metrics for undernourished children and population at risk of hunger give greater emphasis to food security. (Other nutrient

outcomes are not shown.) While rice comes out as the highest-ranked crop under these metrics at the global level (reflecting the scale of its production and consumption), the relative importance of crops differs across metrics and regions. For example, weighting income by the poverty gap index raises the profile of sorghum, millet, yam, and groundnuts (particularly in Sub-Saharan Africa), and reduces that of wheat, potato, and to some extent rice. Other crops are not highly ranked for any of the metrics or regions examined in this study, but might well be ranked more highly for different criteria, locations, or population groups.

## Discussion

This analysis examines the economic impacts of faster crop productivity growth, considering market interactions across multiple commodities and countries, as well as changes in biophysical and socioeconomic factors over time. As such, it offers insights beyond those that can be obtained by considering individual crops or countries in isolation. Nevertheless, it is still a partial perspective addressing specific questions using a particular methodology, and it is important to recognize the limitations inherent in this approach. In this section we comment briefly on the results we found, the methods we used, the process we followed, and implications for further research and decision making.

### Results

We found that increased investment to accelerate crop productivity growth in developing countries can have large impacts on important development indicators. For example, faster productivity growth in rice, wheat and maize was estimated to increase economy-wide income in the selected countries in 2030 by 59 billion USD, 27 billion USD and 21 billion USD respectively (reflecting the scale of their production and consumption), followed by banana and yams with increases of 9 billion USD each. By way of comparison, these amounts are less than 1% of projected GDP in the 106 targeted countries in 2030, but they are 2–15 times current levels of public R&D spending on food crops in developing countries (about 4 billion USD per year, based on estimates from Beintema et al. [50] and ASTI [51]). Income growth was largest in South Asia, but when weighted by poverty measures, the largest increase in income occurred in Sub-Saharan Africa. Faster productivity growth in rice and wheat reduced the population at risk of hunger by 11 million people and 6 million people respectively (representing reductions of 1–2 percent relative to baseline levels in 2030), followed by plantain and cassava with reductions of about 2 million people each. Changes in adequacy ratios for protein and carbohydrates were relatively large, while those for micronutrients were relatively small. As these examples illustrate, the estimated impacts of faster crop productivity growth vary widely across crops, regions, and outcome indicators. This highlights the importance of identifying potentially diverse objectives of different decision makers, recognizing possible tradeoffs between different objectives, and understanding the methods used to generate these results.

### Methods: Models

First, the parity model is relatively simple, intuitive, and well-established, but it focuses on current conditions and historic data. The IMPACT-GLOBE system of models allows exploration of future interactions across crops and countries in the context of changing biophysical and socioeconomic conditions, and is unique among global economic models in covering the 20 crops of interest, but lacks subnational detail (for example, in terms of income classes, rural-urban location, farm size, age, or gender). We note also that GLOBE operates at a different (coarser) level of spatial aggregation than IMPACT, so there is a need to downscale GLOBE results to the IMPACT country level. Downscaling in model ensembles that work at different

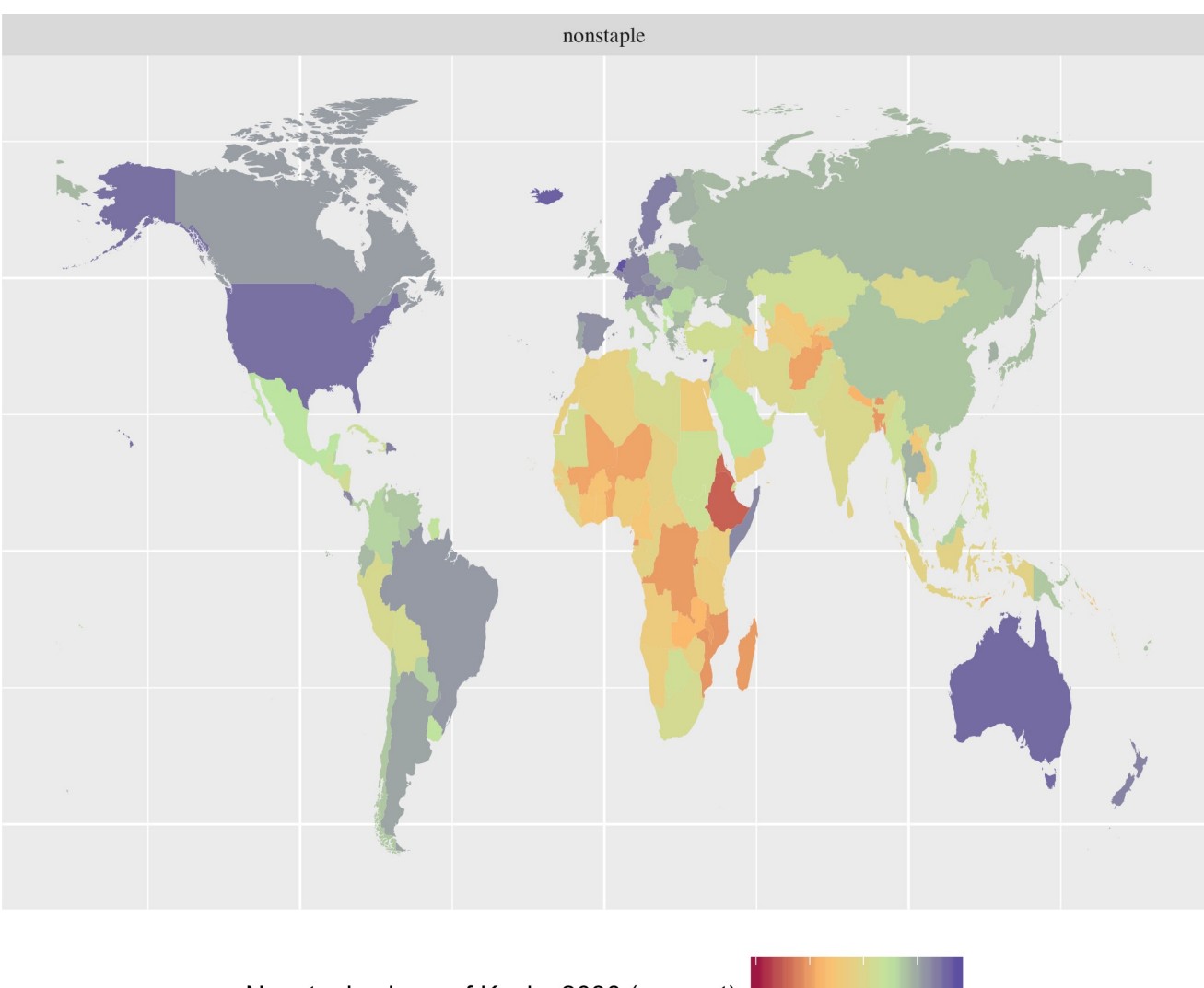

**Fig 4. Non-staple share of dietary energy intake in the reference case in 2030 (percent).** Source: The authors, based on results from the IMPACT model, using a modeling approach detailed in Nelson et al. [45] with Natural Earth map files (https://www.naturalearthdata.com/) using ggplot2 [47] in R [48].

scales is not an exact science and requires additional assumptions. This means that the income results reported here for broad region aggregates are more reliable than the detailed IMPACT country level results (except for cases like China and India, where the GLOBE-IMPACT mapping is 1:1). In relation to nutrient modeling, we note that nutrient availability from crops as estimated here is just one aspect of a more complete characterization of nutrition [45].

## Methods: Assumptions

Second, we considered one specific set of assumptions about changes in population, income and climate (based on SSP2 and RCP8.5). These are standard assumptions by the global modeling community but may not match expectations for particular countries. Different assumptions would generate different results, although the relative economic surplus levels are likely to be robust across the plausible range of parameters. Baseline productivity growth rates

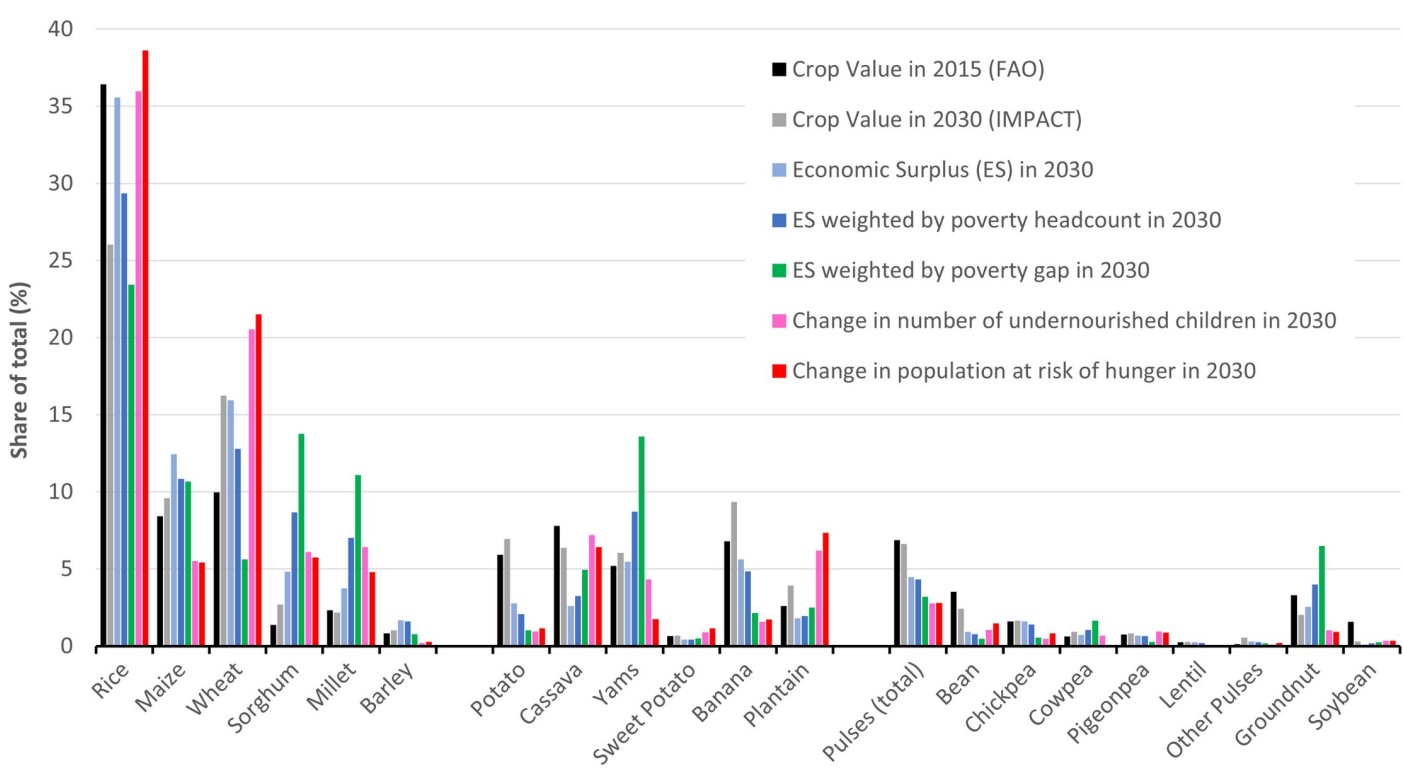

**Fig 5. Relative impacts of faster productivity growth on income, poverty and food security indicators (all 106 countries).** Source: The authors, based on FAOSTAT (2015 production value), IFPRI (IMPACT projections to 2030), PovcalNet (poverty measures, latest available year).

in IMPACT are based on the principles laid out by Evenson and Rosegrant [52] and Evenson et al. [53]. These have been subsequently adjusted based on expert opinion and in consultations with other CGIAR Centers that have expertise on particular crops. However, there is always room for improvement, particularly with minor crops, where knowledge gaps are still greater than with the staple cereals. Assumptions about demand elasticities are important for the consumption side of the modeling presented here. IMPACT's elasticities are originally based on USDA's international database [54] and subsequently adjusted through consultations and feedback from commodity experts in the CGIAR, AgMIP [55], and elsewhere. Alternative elasticities would lead to different outcomes, but again, relative rankings are likely to be similar across the plausible range of elasticities.

## Methods: Scenarios

Third, based on discussion with USAID and the multi-funder group, we focused on stylized scenarios that posit a uniform acceleration of productivity growth for each of 20 selected crops. (Fruits, vegetables, forage crops, and animal source foods were not included in this analysis.) This has the advantage of simplicity and allows comparison of the impacts of a proportionate increase in yields across crops, but is not linked to data or assumptions about costs, rates of technology adoption, or specific investment levels needed to achieve such increases, or how those would likely differ by crop or region. We also examined the impacts of faster productivity growth for each crop individually (while holding productivity growth rates for other crops at their baseline levels), which may have missed potentially interesting interaction effects.

**Table 11. Relative impacts of faster productivity growth on income, poverty and food security indicators: Highest-ranked crops in selected regions.**

| Metric | Region | Highest-ranked crops | Table |
|---|---|---|---|
| Crop Value in 2015 (FAO) | 106 countries | rice, wheat, maize, cassava, pulses, banana | 3 |
| | South Asia | rice, wheat, potato, pulses, banana, maize | 4 |
| | Sub-Saharan Africa | yams, cassava, maize, rice, pulses, groundnuts | 4 |
| Crop Value in 2030 (IMPACT) | 106 countries | rice, wheat, maize, banana, potato, pulses | 3 |
| | South Asia | rice, wheat, potato, banana, pulses, maize | |
| | Sub-Saharan Africa | yams, cassava, maize, plantain, pulses, sorghum | |
| Economic Surplus (ES) in 2030 | Global | rice, wheat, maize, banana, yams, sorghum | 6 |
| | South Asia | rice, wheat, maize, banana, pulses, potato | 7 |
| | Sub-Saharan Africa | rice, maize, sorghum, yams, millet, groundnuts | 7 |
| ES weighted by poverty headcount in 2030 | 106 countries | rice, wheat, maize, yams, sorghum, millet | 6 |
| | South Asia | rice, wheat, maize, banana, pulses, potato | 7 |
| | Sub-Saharan Africa | rice, sorghum, yams, millet, maize, groundnuts | 7 |
| ES weighted by poverty gap in 2030 | 106 countries | rice, sorghum, yams, millet, maize, groundnuts | 6 |
| | South Asia | rice, wheat, maize, banana, pulses, potato | 7 |
| | Sub-Saharan Africa | rice, sorghum, yams, millet, maize, groundnuts | 7 |
| Change in number of undernourished children in 2030 | 106 countries | rice, wheat, cassava, millet, plantain, sorghum | 9 |
| | South Asia | rice, wheat, millet, pulses, maize, chickpeas | |
| | Sub-Saharan Africa | rice, cassava, plantain, sorghum, millet, wheat | |
| Change in population at risk of hunger in 2030 | 106 countries | rice, wheat, plantain, cassava, sorghum, maize | 9 |
| | South Asia | rice, wheat, millet, pulses, maize, chickpeas | |
| | Sub-Saharan Africa | rice, plantain, cassava, wheat, sorghum, maize | |

Note: Pulses include beans, chickpeas, cowpeas, pigeonpeas, lentils, and other pulses (but exclude groundnuts and soybeans). Further details on each metric are provided in Tables 3–9.

Sources: The authors, based on FAOSTAT (2015 production value), IFPRI (IMPACT projections to 2030), PovcalNet (poverty measures, latest available year).

## Methods: Indicators

Fourth, we examined impacts of faster productivity growth on the value of production, economic surplus, nutrient availability and hunger. We did not explore impacts on other

outcomes, including costs of production, net returns, employment, wages, resilience, health or nutrition (including the rise in overweight and obesity among both adults and children, including in developing countries). Lack of data on the costs of achieving faster productivity growth meant we were not able to adequately capture the impact of investments in crops and locations where yields are well below their potential but could be improved relatively cheaply (e.g., through improved management practices) relative to investments in crops and locations that generate high levels of economic surplus but for which further productivity growth might be relatively expensive. Nor did we examine how impacts might vary by gender, age, rural-urban location, farm size, or different weighting schemes. Some of these are within the capabilities of existing models, and some would require further model development, data, or links to other analytical approaches.

## Process

This analysis was demand-led, with specific questions and methods clearly defined and agreed in discussion with the multi-funder group who commissioned the analysis, recognizing the limited time and resources available. Results were shared with the multi-funder group and the CGIAR's Excellence in Breeding Platform in 2018, with two subsequent presentations for clarification and discussion. The analysis was intended to inform dialog and decision making related to the Crops to End Hunger initiative, but we note that the results of this analysis are only one set of inputs to a larger decision-making process that also draws on other analyses and criteria. While intended for a particular audience and purpose, the results may also be of wider interest.

## Implications: Results

As noted above, outcomes for particular crops reflect the scale of their production and consumption, but do not consider the costs of achieving the assumed productivity increases, which may vary significantly across crops and regions. Results of this analysis thus provide an indication of the direction and magnitude of impacts of proportionate increases in productivity growth rates, but can only offer a partial perspective on resource allocation decisions. A more complete perspective would require further analysis with a number of refinements in methods and process.

## Implications: Methods

This type of analysis could be refined and extended in a number of ways to better inform decision making by donor agencies, national governments, and other development partners. First, additional scenarios could be explored, including a wider range of assumptions about socioeconomic and climate pathways, and different scenarios of productivity growth. Second, underlying model parameters such as baseline productivity growth rates and elasticities of supply and demand would benefit from further review and updating. Third, additional crops (such as fruits, vegetables, and forage crops) and animal source foods could be included. Fourth, additional outcome indicators could be examined, including measures of employment, nutrition, health, greenhouse gas emissions and other environmental indicators. Fifth, targeted model improvements would allow analysis of sub-national variations in outcomes for different population groups, including by income, gender, age, or rural-urban location. Sixth, improved data on the costs of technology development and dissemination–including both public and private R&D–would allow estimation of rates of return in addition to impacts of alternative policy and investment options. And seventh, valuable extensions of this analysis could include estimating differential elasticities of productivity growth with respect to investments in crop

breeding or improved management for different types of crops. This could also be supplemented by expert opinion on the probability of success of crop breeding or improved management for different crops relative to expenditures on these crops. This additional analysis could be embedded in the modeling or could be used by decision-makers to further inform investment decisions, taking account of the probability of success and the cost per unit of productivity gain and the resulting impacts on key indicators.

## Implications: Process

In addition to improvements in analytical methods, our experience also suggests several ways in which the process of analysis to inform decision making could be improved. First, the more attention that can be given early in the process to identifying relevant stakeholders (including funders, researchers, decision makers, and others who will be affected by any resulting decisions) along with their various interests and questions, the better focused the analysis can be and the more useful its results. Second, to enhance the transparency, technical quality and credibility of future work, it would be beneficial to develop a systematic process to review and update the models and parameters used in this analysis on an on-going basis, in collaboration with experts across the CGIAR and beyond. Third, to increase relevance, understanding, and confidence in results, it would be helpful to establish an on-going process for iteration between users and providers of this type of analysis to allow for dialog on results, discussion, revisions, new questions, and further analysis. Finally, we recognize that these steps are costly and resources are limited, but we believe that investment in a systematic and iterative process would generate improvements in the speed, transparency, quality, credibility and relevance of future analysis to inform decision making in this area.

## Acknowledgments

This paper builds on a project note originally delivered to USAID by the authors in June 2018, parts of which were subsequently summarized in a CGIAR System Council document in October 2018 [56]. Feedback from partners in the Crops to End Hunger initiative is gratefully acknowledged. We also thank Olaf Erenstein, Gideon Kruseman, and four anonymous reviewers for their helpful comments. The views and findings presented here are those of the authors, and may not be attributed to IFPRI, ERS, USAID or other affiliated institutions.

## Author Contributions

**Conceptualization:** Keith Wiebe, Timothy B. Sulser, Mark W. Rosegrant, Keith Fuglie, Dirk Willenbockel, Gerald C. Nelson.

**Formal analysis:** Keith Wiebe, Timothy B. Sulser, Shahnila Dunston, Mark W. Rosegrant, Keith Fuglie, Dirk Willenbockel, Gerald C. Nelson.

**Funding acquisition:** Mark W. Rosegrant.

**Investigation:** Keith Wiebe, Timothy B. Sulser, Shahnila Dunston, Mark W. Rosegrant, Keith Fuglie, Dirk Willenbockel, Gerald C. Nelson.

**Methodology:** Keith Wiebe, Timothy B. Sulser, Shahnila Dunston, Mark W. Rosegrant, Keith Fuglie, Dirk Willenbockel, Gerald C. Nelson.

**Project administration:** Keith Wiebe.

**Supervision:** Keith Wiebe.

**Visualization:** Keith Wiebe, Timothy B. Sulser, Keith Fuglie, Gerald C. Nelson.

**Writing – original draft:** Keith Wiebe, Timothy B. Sulser, Mark W. Rosegrant, Keith Fuglie, Dirk Willenbockel, Gerald C. Nelson.

**Writing – review & editing:** Keith Wiebe, Timothy B. Sulser, Shahnila Dunston, Mark W. Rosegrant, Keith Fuglie, Dirk Willenbockel, Gerald C. Nelson.

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
