## [Decision Letter · Decision Letter 0]

23 Jun 2020

PONE-D-20-06988

Modeling impacts of faster crop productivity growth to inform the CGIAR initiative on Crops to End Hunger

PLOS ONE

Dear Dr. Wiebe,

Thank you for submitting your manuscript to PLOS ONE. After careful consideration, we feel that it has merit but does not fully meet PLOS ONE’s publication criteria as it currently stands. Therefore, we invite you to submit a revised version of the manuscript that addresses the points raised during the review process.

Making assumptions explicit and explaining the rationale for making assumptions that have a direct impact om key outcomes is essential for any modeling exercise and is reiterated by the reviewers of the manuscript. A better justification of assumptions will be required.

It is also essential to make all the underlying data and model results publicly available. Not only is this a prerequisite for publication of a manuscript in PLOSone, this also facilitates peer review of results and helps identify inadvertent errors. 

Feel free to reach out to discuss any questions you may have on how to address the comments of the reviewers.

We look forward to receiving your revised manuscript.

Kind regards,

Gideon Kruseman, Ph.D.

Academic Editor

PLOS ONE

Additional Editor Comments:

Making assumptions explicit and explaining the rationale for making assumptions that have a direct impact om key outcomes is essential for any modeling exercise and is reiterated by the reviewers of the manuscript. 

One way of addressing some of the concerns raised is adding through supplementary documentation the sensitivity analysis results related to contested parameter choices.

It is also essential to make all the underlying data and model results publicly available. Not only is this a prerequisite for publication of a manuscript in PLOSone, this also facilitates peer review of results and helps identify inadvertent errors. The prepublication of the manuscript on SocArXiv [osf.io/preprints/socarxiv/h2g6r] brought a calculation error to the fore that you have since remedied.

Looking forward to the revised version of the manuscript addressing the important comments by the reviewers.

Journal Requirements:

3. We note that Figures 2, 3a, 3b and 4 in your submission contain map images which may be copyrighted. All PLOS content is published under the Creative Commons Attribution License (CC BY 4.0), which means that the manuscript, images, and Supporting Information files will be freely available online, and any third party is permitted to access, download, copy, distribute, and use these materials in any way, even commercially, with proper attribution. For these reasons, we cannot publish previously copyrighted maps or satellite images created using proprietary data, such as Google software (Google Maps, Street View, and Earth). For more information, see our copyright guidelines: http://journals.plos.org/plosone/s/licenses-and-copyright.

3.1.    You may seek permission from the original copyright holder of Figures 2, 3a, 3b and 4 to publish the content specifically under the CC BY 4.0 license.

3.2.    If you are unable to obtain permission from the original copyright holder to publish these figures under the CC BY 4.0 license or if the copyright holder’s requirements are incompatible with the CC BY 4.0 license, please either i) remove the figure or ii) supply a replacement figure that complies with the CC BY 4.0 license. Please check copyright information on all replacement figures and update the figure caption with source information. If applicable, please specify in the figure caption text when a figure is similar but not identical to the original image and is therefore for illustrative purposes only.

Reviewers' comments:

Reviewer's Responses to Questions

**Comments to the Author**

1. Is the manuscript technically sound, and do the data support the conclusions?

Reviewer #1: Yes

Reviewer #2: Yes

Reviewer #3: Partly

2. Has the statistical analysis been performed appropriately and rigorously? 

Reviewer #1: Yes

Reviewer #2: N/A

Reviewer #3: Yes

3. Have the authors made all data underlying the findings in their manuscript fully available?

Reviewer #1: No

Reviewer #2: No

Reviewer #3: Yes

4. Is the manuscript presented in an intelligible fashion and written in standard English?

Reviewer #1: Yes

Reviewer #2: Yes

Reviewer #3: Yes

5. Review Comments to the Author

Reviewer #1: This is a sound paper that addresses an important issue. It is ambitious and sometimes goes beyond the analytical capacity of the models.

Major comments

This is a sound analysis of an important problem—how to determine priorities among many CGIAR crops.

The discussion covers a number of limitations of the analysis but there are others as well. Also quality of data need to be addressed. Finally, the conclusions should be sharpened. See details below.

1. The high value of yams and bananas deserves more discussion. How are FAOSTAT prices for these crops determined. Are they farmgate prices or market prices that include high transport margins?

2. No distinction is made between food and feed uses of crops. Clearly, feed uses contribute less to energy and would favor richer consumers.

3. The analysis is for the public sector only. We know that for some crops, maize and potatoes, for example, private R&D is important and growing.

4. Table 7 on rankings deserves more discussion and sharper conclusions. Rice is clearly the number one priority in all cases. Wheat is also ranked high under most assumptions. At the same time, some crops are clearly low priority under all assumptions—barley and several of the legumes. The conclusions should clearly state these outliers and also that more data and more in depth analysis would be unlikely to change these conclusions.

Other comments

44. Needs to specify the base year

138. this is a big assumption and deserves more attention in the discussion.

Table 2. No rationale is given for the selection of countries. China and Brazil are not included but India is.

Reviewer #2: The authors conduct an interesting analysis about priorities and benefits of agricultural productivity enhancements. There are many things I like about the analysis. However, the following comments will focus on potential weaknesses or shortcomings. I spend about 3 hours reading the paper. If I misinterpret some things, the authors should not hesitate to correct me.

My main criticism is that the analysis seems somewhat incomplete and inconsistent.

1) The link between IMPACT and GLOBE seems to be a single feedback loop. I did not see a statement that this suffices to achieve consistency between the two models. It would find it sensible to iterate information between the two models until consistency is achieved. I don't think that it would take a lot of iterations but it would be useful to have some information about this in an appendix. Especially in poorer countries where agriculture contributes substantially to GDP, a single feedback may still leave biases.

2) Parity analysis seems to provide only a demand value for productivity improvements. However, the cost of productivity improvements may differ greatly. For example, if a crop is not managed well in a certain region, parity analysis will suggest a lower priority (resulting from lower production quantities). However, the cost of productivity improvement may be much lower for low-yield situations (large yield gaps) than for crops which are already well managed and closer to some natural limits. The authors state clearly that they do not look into the cost of productivity improvements. However, I think they should at least discuss the issue of the yield gap a little bit better to avoid a misinterpretation of their results. Again, all other things equal, lower existing yields would with the applied method result in lower research priorities. I think, the lower existing yields (higher yield gaps) increase the attractiveness for research spending.

To remedy this, the authors could use two alternative accounting schemes: a) price x area -> this switches off the yield impact, b) price x area x yield gap -> this would give priority to crops with higher yield gaps. The yield gap, ideally, should be calculated taking into account the regional distribution of soil-climate conditions. I believe there are already several studies which estimate regionally resolved yield gaps at global level.

3) The authors estimate research spending priorities based on a single independent experiment for each crop where they raise the productivity of this crop by 25% (I believe globally). It would be very interesting to see what happens when a certain research budget is allocated simultaneously to regions and crops according to the estimated priorities. This would truly make the analysis complete and would consider the interactions between the crops. Then, they could trace out a response function where regionally diverse benefits of productivity increases are estimated against the regionally and crop specific expenses for different levels of budget. This, in my opinion would make the analysis really useful to policy makers.

Other comments

The poverty weighting schemes are good in principal but a bit crude. The crudeness comes from using the 1.90 $/day value as a black and white discrimination between poor and not poor. I would find it more sensible, if the authors would use an integral under an income distribution function. However, this would then also need additional assumptions on how to weight levels of poverty.

The authors state that the results data will be freely available and accessible. However, if I understand the policy of the journal correctly, they should also disclose the input data of the model.

Where is the information referred to in these statements:

".. baseline rates of productivity growth assumed in the IMPACT"

"IMPACT uses assumptions about key drivers such as population, income, technology, policy and climate to simulate changes in agricultural demand, production and markets for 60 commodities in 158 countries"

Reviewer #3: This paper is a follow-up of a study carried out by IFPRI and USDA-ERS for USAID to inform the Crops to End Hunger (CtEH) initiative. The latter is a development aid program which seeks to modernize public plant breeding in lower-income countries. The IFPRI-ERS study was aimed at assessing the impacts of faster productivity growth for selected food crops on income and other key indicators in developing countries in 2030.

The paper describes the method used and the main findings of this study. The method used is not original: this is a simulation exercise of a +25% increase of crop yields (crop by crop) in developing countries, carried out with the IMPACT model coupled with the GLOBE model (this coupling is already used in Mason-D’Croz et al., 2019, World Development); outputs are the impacts on economic surplus, prevalence of hunger and availability and adequacy of key nutrients (like in Nelson et al., 2018, Nature Sustainability). Results are unsurprising: as the yield increase applied successively to each crop is of the same magnitude (+25% relative to the baseline), the most important crops in terms of production value induce the strongest effects in terms of income and the most important crops in the diets induce the highest effects on food and nutritional security.

The paper is well-written and reports a competent work. But, I wonder what is the actual value added of the paper. Linked to this question, I have several concerns that I describe below.

1. Major issues

i) I don’t understand why the authors refer to the parity model. This is very confusing and, as far as I can understand, it adds nothing to the analysis.

It is very confusing because, as mentioned by the authors, the parity model is commonly used to allocate R&D resources in multi-commodity systems. So initially the reader may imagine that the budget of the CtEH is allocated among the different crops in a first stage, that, in the second stage, the study estimates the impacts of additional R&D efforts on each crop yield and that, in the last stage, the study examines the impact of the induced differentiated yield changes in terms of income and food and nutritional security. In fact the study does not do that at all and the starting point is not the parity model but the assumed +25% yield increase for all crops. This +25% yield increase is simulated crop by crop and the induced impact in terms of income and food and nutritional security indicators are compared and analysed. Maybe I’ve missed something but in such approach, I don’t see what the parity model actually adds.

We can imagine that the parity model analysis could indicate which crops should receive higher shares of R&D resources. Then the simulation results would inform on which crops induce the highest income and food and nutritional effects following a x% increase in its yield. And finally we could examine whether the most efficient crops (in terms of increasing income and food and nutritional security) are also the ones which should receive higher shares of R&D resources according to the parity model (using alternative rules through poverty weights). But, once again as far as I understand, this is not was is done neither, otherwise authors would not state that “The scenarios of accelerated productivity growth reported in this note assume that investment in new varieties and other sources of on-farm productivity growth will increase sufficiently to result in a 25% increase in the annual rate of yield growth above “baseline” yield growth in farmers’ fields over the period 2015-2030” (li 77 to 80) or “A third departure from the simple parity model is to determine how accelerated crop yield growth might affect future incomes” (li 155-156).

Once again this is very confusing: are we in the first approach (impacts of R&D on yields, and impacts of yield changes on income and food and nutrition security), but in this case why a uniform +25% yield increase? Or are we in the second approach (impacts of yield changes on income and food and nutritional security and comparison with recommandations of R&D resource allocation according to the parity model), but in this case why justifying the 25% yield increase as resulting from additional R&D resources?

ii) If the +25% yield increase results from additional R&D resources (from theCtEH I guess?), the paper should explain what are the bases of this assumption (empirical, literature, expert knowledge?). This is not done at all in the paper (which makes me think that we are rather in the second approach which does not require to explain the retained level of yield increase. This level could be 1%, 10%, 20%, etc., applying the same percentage to all crops being the only requirement in this case).

iii) If I can understand that the authors use poverty rates to weight production values in the parity model I don’t understand why the same weights are applied to the simulated economic surplus from the accelerated productivity growth scenarios.

Once again if we are in the second approach, simulation results should be used as an estimation of the relative efficiency of each crop in increasing income and food and nutritional security and then should be compared to recommendations resulting from the parity model. The use of poverty weights makes sense in the parity model since it allows to show how the R&D resource allocation should change if we want to put emphasis on the poorest population. But, as far as I can understand, applying poverty weights to the simulation results of the accelerated productivity growth scenarios makes no sense in this case.

iv) Choosing clearly between the first or the second approach would provide a more clear grid for analysing results and highlighting the main findings, which is not the case currently.

Currently the analysis of scenarios’ simulation results are a mix between what it would be under the first option and what it would be under the second option. As a result, it is very difficult for the reader to see what these results really add.

Minor issues

i) Introduction

- The paper should be situated relative to the existing literature and its originality, contribution and value added regarding existing knowledge should be described.

- Li 48 to 51. Changing diets and reducing food waste have also been advocated as part of the solution. This should be mentioned and related literature added.

- Li 65-66: “This paper briefly describes how we did that and what we found”: this looks like a summary of a study report not like a research paper. This relates to the above-described major issues: what is the research question authors are dealing with? What do they want to show? This should be specified.

ii) Approach

- It should be made clear for what purpose the parity model is used (see major issues, first or second approach).

- Li 77-80: It should be made clear why the authors retained scenarios involving a uniform +25% yield increase? For what purpose such scenarios have been defined?

iii) Analysis using the parity model

- Li 127-128: Explain why production quantities averaged over 2014-2016 are valued at international prices averaged over 2004-2006.

iv) Scenarios of faster productivity growth

- Li 155-156: I don’t understand this sentence.

- Explain the 25% level of yield increase. Explain whether there is a link between these +25% and additional R&D resources from the CtEH program. Explain whether there is a link between these +25% and the parity model. The current version of the text is very confusing and it is difficult to understand where the +25% are coming from and what are they dedicated to show?

v) Analysis using the IMPACT model

- Li 198-202: the iteration process between IMPACT and GLOBE should be described in more details. Only one iteration is mentioned: from IMPACT to GLOBE and then from GLOBE to IMPACT. Does it mean that only one iteration is sufficient to reach a consistent global equilibrium in both models?

vi) Analysis using the GLOBE model

- Li 222-225: the way the “agricultural productivity enhancement from the various scenarios … in IMPACT” are translated into “shifts of the factor productivity parameters in the agricultural production functions” should be described in more details. Are these productivity shifts also affecting land rents (only wages and capital returns are mentioned)?

vii) Results

- Li 348-350: “Growth in unweighted economy-wide income was largest in South Asia, but when weighted by the poverty gap, the largest increase occurred in Sub-Saharan Africa.”. Maybe “occurred” is not the right verb.

- Table 4a: instead of totals computed as the sum of the values obtained from each crop scenario, a “true” total reporting the effect of a +25% yield increase for all crops considered simultaneously would bring additional information on the interaction between crop markets and cross effects of additional R&D resources allocated to various crops.

- Table 4c: Could be moved to additional information or appendix.

6. PLOS authors have the option to publish the peer review history of their article (what does this mean?). If published, this will include your full peer review and any attached files.

Reviewer #1: No

Reviewer #2: No

Reviewer #3: No

---

## [Author Response · Author response to Decision Letter 0]

19 Sep 2020

Responses to the reviewers' comments have been provided in a separate file.

---

## [Decision Letter · Decision Letter 1]

27 Oct 2020

PONE-D-20-06988R1

Modeling impacts of faster productivity growth to inform the CGIAR initiative on Crops to End Hunger

PLOS ONE

Dear Dr. Wiebe,

Thank you for submitting your manuscript to PLOS ONE. After careful consideration, we feel that it has merit but does not fully meet PLOS ONE’s publication criteria as it currently stands. Therefore, we invite you to submit a revised version of the manuscript that addresses the points raised during the review process.

You may notice we have added a new reviewer. We chose this reviewer from a set of scholars that had provided some feedback to me regarding the prepublication version on SocArXiv (https://doi.org/10.31235/osf.io/h2g6r). The remarks provided by this scholar are in line with the remarks by the other reviewers, but provide suggestions that can benefit the revision.

We look forward to receiving your revised manuscript.

Kind regards,

Gideon Kruseman, Ph.D.

Academic Editor

PLOS ONE

Additional Editor Comments (if provided):

The fundamental issue with any paper based on a modeling exercise that may be used beyond the bubble of modelers that know and understand the inner workings of a model is to provide sufficient context to allow the model outcome analysis results to be used in a meaningful way. The results presented in the paper will be variously used without necessarily a full understanding of the limitations.

This requires some modesty is presentation. It also requires providing clear delimitation of scope and related to that what key assumptions have been made at the model level and at the scenario level. The best way to do this is to provide adequate supplementary material spelling out explicitly assumptions, data sources, standards and definitions used.

Providing (a link to) the data used in the analysis is a prerequisite for publication in PLOS One anyway. Not all data are easily accessible.

There are numerous other papers using the same suite of models by largely the same set of authors. This paper potentially adds value but needs to be more transparent about some of the issues raised by the reviewers. In any case, the paper is already available in SocArXiv (https://doi.org/10.31235/osf.io/h2g6r) so this version when (/if) published should try to address remaining issues for the benefit of all.

Reviewers' comments:

Reviewer's Responses to Questions

**Comments to the Author**

1. If the authors have adequately addressed your comments raised in a previous round of review and you feel that this manuscript is now acceptable for publication, you may indicate that here to bypass the “Comments to the Author” section, enter your conflict of interest statement in the “Confidential to Editor” section, and submit your "Accept" recommendation.

Reviewer #1: (No Response)

Reviewer #3: (No Response)

Reviewer #4: (No Response)

2. Is the manuscript technically sound, and do the data support the conclusions?

Reviewer #1: Yes

Reviewer #3: Yes

Reviewer #4: Partly

3. Has the statistical analysis been performed appropriately and rigorously? 

Reviewer #1: Yes

Reviewer #3: Yes

Reviewer #4: N/A

4. Have the authors made all data underlying the findings in their manuscript fully available?

Reviewer #1: Yes

Reviewer #3: Yes

Reviewer #4: No

5. Is the manuscript presented in an intelligible fashion and written in standard English?

Reviewer #1: Yes

Reviewer #3: Yes

Reviewer #4: Yes

6. Review Comments to the Author

Reviewer #1: (No Response)

Reviewer #3: Thank you for your point-by-point responses to my first round comments. I agree with most of your responses but I still have one concern that I detail below.

I think I’ve finally found why I had difficulties to connect the parity model approach and the IMPACT-GLOBE simulation approach proposed here: the production value share indicator used in the parity model approach is very different from the ES value share indicator used in the IMPACT-GLOBE simulation approach. And I even wonder if both indicators can be compared, from an economic point of view.

Therefore, I think the authors should explain the exact meaning of their ES (economic surplus) share indicator. When dealing with the value shares, we refer to an initial situation (FAO data 2015) or to a single scenario (IMPACT model 2015 or IMPACT model 2030 reference scenario). In that case, the value share of each crop is the production value of that crop divided by the sum of all crop production values in the same situation or scenario.

When dealing with the ES shares, the situation is very different. If I understand well the so-called Economic surplus (ES) of each crop (Tab. 4a) is the welfare change induced by the +25% productivity improvement simulated for that crop relative to the reference scenario. Thus the ES of each crop results from different scenarios (ES rice is the welfare change resulting from the rice productivity improvement scenario, the maize ES is the welfare change resulting from the maize productivity improvement scenario and so on). In such a case, what is the meaning of the sum of ES over all crops (the sum of the welfare changes obtained from the individual crop productivity improvement scenarios, which is clearly different from the welfare change resulting from the all crops productivity improvement scenario)? Furthermore, what is the meaning of the ES share of each crop? Once again, I am not sure I understood correctly, but the ES share of one crop is the welfare change resulting from that crop productivity improvement scenario divided by the sum of the welfare changes resulting from all individual crop productivity improvement scenarios. I am not sure I get exactly what does this ES share indicator mean. Is this indicator consistent from an economic point of view? Can it be compared with the production value share indicator used in the parity model? It would be consistent and comparable to the production value share indicator if the ES share of one crop was the welfare change resulting from that crop productivity improvement scenario divided by the welfare change resulting from the scenario involving simultaneous productivity improvement of all crops (the ES share of one crop would measure the contribution of that crop productivity improvement to the global welfare change obtained from the productivity enhancement of all crops).

Maybe I’ve missed something and I’ll be happy to get some explanation from the authors. But if I understood correctly, at least this point should be made clear to the reader and implications should be discussed.

Reviewer #4: Review PONE-D-20-06988R1

The revised paper has been variously updated in response to reviewer comments. Overall, this has improved and clarified a number of issues raised. Yet a number of issues remain. Given the complexity of the modeling and associated assumptions and implications of the results do see the need to address these issues clearly. Some readers may use the quantitative results at face value – especially as many will not understand or easily find the many underlying and sometimes substantial assumptions.

Some major issues:

1. Crop prices used: The paper uses “global average commodity prices from 2004-06 (i.e., in constant 2005 international dollars)”. Various reviewers took issue with these outdated prices and the authors did add but not resolve the issue. The paper and Tables value the different commodities using FAO 2014-16 production data and these 2004-06 average prices (a decade earlier!). Tables and scenarios always refer to “2015” (e.g. Table headings: “Gross production value from FAOSTAT in 2015”), yet using outdated prices which is then mentioned in the footnotes. This is not only confusing but also particularly problematic:

a. As prices are 1 of the 2 key factors determining the relative commodity values in the baseline year.

b. The world has had a global food crisis during this interlaying decade with prices for some of these food crops showing significant movements and being at the heart of the crisis. It seems rather simplistic to just use the outdated prices and assume that there may not have been any relative shift in prices since.

c. The models and scenarios subsequently project to 2030. For such 15 year projections one would expect to use indicators/trends as close to the 2015 base year as possible – and not add another decade of noise and potential bias with outdated prices …

The authors add some text (incl that FAO uses the Geary-Khamis method to derive a set of average global commodity prices in purchasing-power-parity dollars per metric ton) and refers to FAOStat . The text states “no such internationally comparable average producer prices exist for a more recent period” – yet the referred to FAOStat does includes more recent price data. Further effort should be made to really use 2015 as a base year for projections – not as an ambiguous and confounding base year that combines prices for a decade earlier (2004-06!) with production data from 2015.

2. Clarifying critical assumptions in additional tables: One could argue over a number of the underlying assumptions. But much of the models remain a black box – with limited insight in what is actually happening. In the responses authors mention uploading the data to GitHub and providing the relevant access information prior to publication. But one would expect to at least be able to access some of the critical assumptions before deciding on publication and the utility of the paper – also for subsequent users. The paper is pretty long – but it would not hurt to add a few tables with the critical assumptions as they apply to each of the studied crops – in the supplementary data section if preferred. The authors do variously mention the intention to improve on these results in the discussion – why not then at least share the underlying key assumptions …? For instance:

a. Baseline productivity growth rates assumed in the IMPACT model: The paper states the authors use these base rates and then increase these with a flat 25% and project to 2030. In the current version these basic historic rates are a key driver for 2030 impacts and would be good to clarify and make these underlying rates clearly accessible or point to full details if available elsewhere … Now such statements on p. 34 as “Baseline productivity growth rates in IMPACT are based on the principles laid out” in the literature. “These have been subsequently adjusted based on expert opinion and in consultations”. “However, there is always room for improvement”. If the underlying growth rate was negligible – 25% increase may not do much. Please just be specific and transparent and list them out.

b. p. 34-5 “Demand elasticities are important”. “Alternative elasticities would lead to different outcomes, but again, relative rankings are likely to be similar across the plausible range of elasticities.” Again, please just be specific and transparent and list them out.

3. Avoiding ambiguous indicators:

a. Clarifying the base reference: In a number of instances the 2030 implications are presented as changes relative to a base – but no actual base reference numbers are provided to at least gauge the relative magnitude. A clear example is Table 5. The proposed scenarios would lift a potential aggregate of 27.5M out of hunger relative to the “reference scenario in 2030”. So what was the projected hunger rate in 2030 as well as the current/base rate? Same for undernourished children.

b. “Adequacy ratios”: Fig 3 presents adequacy ratios for the reference case in 2030 for selected nutrients and vitamins. For a number of countries some adequacy ratios exceed 3, and for carbohydrates this is nearly global. This seems high vis-à-vis concerns of feeding the world. It also contradicts earlier published findings by many of the same authors using the same underlying models. E.g. Mason-D’Croz et al., 2019, World Development Fig 8 depicts average food supply (kilocalorie per person per day) in 2010 and 2030 with global average supply typically approaching the recommended daily consumption of an active 20 to 35-year-old male. Please cross-check what is going on and update as needed. Suggest to drop Fig 3.

c. “dietary diversity”: p. 30 states “Several measures of diversity are available. For this report, we use the non-staple share of energy intake.” This is a rather questionable choice. Dietary diversity primarily relates to diverse food groups to ensure adequate intake across nutrients. Staples are a good and generally cheaper source of energy – diversity is needed to ensure adequacy in relation to other nutrients. Also questions arise around what the authors define as “staples”. E.g. surprised to note that banana’s are not considered staples: “The only yield increases that raise the non-staple share more than 0.01 percent are for groundnuts in SSA (0.08 percent) and banana in Asia and SSA (0.03 and 0.04 respectively)”. Also questions about what is actually included in the “nonstaple” share of kCals in much of the Global North (>60%). Drop Fig 4 given this ambiguity.

d. Fig 5 has a rather unclear legend for the metrics presented as they represent different years (2015, 2030) and scope (some seemingly share in the total metric for the year – others like undernourished children and hunger share in the change of the metric?). Crop value clearly relates to 2 different years/estimates; but ES – to which year does this relate (2030?). Also reference to “shocks” in the title – whereas ones assumes this relates to the assumed 25% productivity increase for each crop? Table 7 – same issue with metrics as Fig 5.

7. PLOS authors have the option to publish the peer review history of their article (what does this mean?). If published, this will include your full peer review and any attached files.

Reviewer #1: No

Reviewer #3: No

Reviewer #4: No

---

## [Author Response · Author response to Decision Letter 1]

11 Jan 2021

Responses to the reviewers' comments have been attached in a separate file.

---

## [Decision Letter · Decision Letter 2]

9 Feb 2021

PONE-D-20-06988R2

Modeling impacts of faster productivity growth to inform the CGIAR initiative on Crops to End Hunger

PLOS ONE

Dear Dr. Wiebe, dear Keith

Thank you for resubmitting your manuscript to PLOS ONE. After careful consideration, we feel that it is almost there but requires some finishing touches. Therefore, we invite you to submit a revised version of the manuscript that addresses the final three points raised during the review process.

Reviewer number 3 has a valid point about the note with Table 4. Please clarify.Value based analyses depend critically on prices used. Prices are notoriously tricky values in global statistics because there is not always clarity about what product the price refers to. For instance roots and tubers are harvested with high moisture content reflected in production statistics, prices sometimes refer to the wet product and sometimes to dried product. The overly high values of for instance cassava in west Africa can be attributed to this issue. Without delving into details, a note of caution about values is warranted. The Geary-Khamis method does not solve this. This implies a word of caution about the statement in lines 390-393. The weighing scheme based on $1.90/day/capita poverty rate [24] see line 173 assumes comparable data. Not all countries are covered and the data for different countries come from different years. A note of caution is therefore warranted besides the warning in line 221.

We look forward to receiving your revised manuscript.

Kind regards,

Gideon Kruseman, Ph.D.

Academic Editor

PLOS ONE

Additional Editor Comments (if provided):

A word of caution is warranted about the data used that may pose some problems. This was raised earlier in the review process as well. two examples stand out:

weighing with poverty rates is not completely unproblematic, when this data is missing for some countries.Prices for roots and tubers sometimes refer to fresh produce and sometimes to dried commodities. For instance for cassava in west Africa this is an issue.

Addressing these data issues always has an arbitrary component to it. That is not a problem as long as you are transparent about this.

Reviewers' comments:

Reviewer's Responses to Questions

**Comments to the Author**

1. If the authors have adequately addressed your comments raised in a previous round of review and you feel that this manuscript is now acceptable for publication, you may indicate that here to bypass the “Comments to the Author” section, enter your conflict of interest statement in the “Confidential to Editor” section, and submit your "Accept" recommendation.

Reviewer #1: All comments have been addressed

Reviewer #3: (No Response)

2. Is the manuscript technically sound, and do the data support the conclusions?

Reviewer #1: Yes

Reviewer #3: Yes

3. Has the statistical analysis been performed appropriately and rigorously? 

Reviewer #1: Yes

Reviewer #3: Yes

4. Have the authors made all data underlying the findings in their manuscript fully available?

Reviewer #1: Yes

Reviewer #3: Yes

5. Is the manuscript presented in an intelligible fashion and written in standard English?

Reviewer #1: Yes

Reviewer #3: Yes

6. Review Comments to the Author

Reviewer #1: (No Response)

Reviewer #3: I think the explanation added as a note to Table 4 (a, b and c)( for ease of comparison ...) is not useful and may be confusing. What I requested was a clear explanation that ESS=ES share = ES cropi/Sum over cropi (EScropi) and that EScropi is a different scenario for each cropi. So that ES crop1 is Economic surplus for scenario1 and ES share crop1= ES scenario1/ES scenario1+ES scenario2+ES scenario3+...

In addition if you decide to keep the sentence "For ease of comparison ..." It should come after the definition of ESS. This is not the case currently.

7. PLOS authors have the option to publish the peer review history of their article (what does this mean?). If published, this will include your full peer review and any attached files.

Reviewer #1: No

Reviewer #3: No

---

## [Author Response · Author response to Decision Letter 2]

26 Mar 2021

Responses to the reviewers' comments have been attached in a separate file.

---

## [Editor Report · Decision Letter 3]

30 Mar 2021

Modeling impacts of faster productivity growth to inform the CGIAR initiative on Crops to End Hunger

PONE-D-20-06988R3

Dear Dr. Wiebe,

We’re pleased to inform you that your manuscript has been judged scientifically suitable for publication and will be formally accepted for publication once it meets all outstanding technical requirements.

Kind regards,

Gideon Kruseman, Ph.D.

Academic Editor

PLOS ONE

Additional Editor Comments (optional):

Note that reference 29 states "in press" while it has been published since, also the title is slightly different. That will need to be rectified before publication. https://acsess.onlinelibrary.wiley.com/doi/full/10.1002/csc2.20114
---

## [Editor Report · Acceptance letter]

5 Apr 2021

PONE-D-20-06988R3 

Modeling impacts of faster productivity growth to inform the CGIAR initiative on Crops to End Hunger 

Dear Dr. Wiebe:

I'm pleased to inform you that your manuscript has been deemed suitable for publication in PLOS ONE. Congratulations! Your manuscript is now with our production department. 

Kind regards, 

on behalf of

Dr. Gideon Kruseman 

Academic Editor

PLOS ONE